# Instance-Dependent Bounds for Zeroth-order Lipschitz Optimization with Error Certificates

**François Bachoc**
Institut de Mathématiques de Toulouse & University Paul Sabatier
`francois.bachoc@math.univ-toulouse.fr`

**Tommaso Cesari**
Toulouse School of Economics
`tommaso-renato.cesari@univ-toulouse.fr`

**Sébastien Gerchinovitz**
IRT Saint Exupéry & Institut de Mathématiques de Toulouse
`sebastien.gerchinovitz@irt-saintexupery.com`

## Abstract

We study the problem of zeroth-order (black-box) optimization of a Lipschitz function $f$ defined on a compact subset $\mathcal{X}$ of $\mathbb{R}^d$, with the additional constraint that algorithms must certify the accuracy of their recommendations. We characterize the optimal number of evaluations of any Lipschitz function $f$ to find and certify an approximate maximizer of $f$ at accuracy $\varepsilon$. Under a weak assumption on $\mathcal{X}$, this optimal sample complexity is shown to be nearly proportional to the integral $\int_{\mathcal{X}} d\boldsymbol{x}/(\max(f) - f(\boldsymbol{x}) + \varepsilon)^d$. This result, which was only (and partially) known in dimension $d = 1$, solves an open problem dating back to 1991. In terms of techniques, our upper bound relies on a packing bound by Bouttier et al. (2020) for the Piyavskii-Shubert algorithm that we link to the above integral. We also show that a certified version of the computationally tractable DOO algorithm matches these packing and integral bounds. Our instance-dependent lower bound differs from traditional worst-case lower bounds in the Lipschitz setting and relies on a local worst-case analysis that could likely prove useful for other learning tasks.

## 1 Introduction

The problem of optimizing a black-box function $f$ with as few evaluations of $f$ as possible arises in many scientific and industrial fields such as computer experiments (Jones et al., 1998; Richet et al., 2013) or automatic selection of hyperparameters in machine learning (Bergstra et al., 2011). For safety-critical applications, e.g., in aircraft or nuclear engineering, using sample-efficient methods is not enough. *Certifying* the accuracy of the output of the optimization method can be a crucial additional requirement (Vanaret et al., 2013). As a concrete example, Azzimonti et al. (2021) describe a black-box function in nuclear engineering whose output is a $k$-effective multiplication factor, for which a higher value corresponds to a higher nuclear hazard. Certifying the optimization error is a way to certify the worst-case $k$-effective factor, which may be required by safety authorities.

In this paper, we formally study the problem of finding and certifying an $\varepsilon$-approximate maximizer of a Lipschitz function $f$ of $d$ variables and characterize the optimal number of evaluations of any such function $f$ to achieve this goal. We start by formally defining the setting.

35th Conference on Neural Information Processing Systems (NeurIPS 2021).

## 1.1 Setting: Zeroth-order Lipschitz Optimization with Error Certificates

Let $f \colon \mathcal{X} \to \mathbb{R}$ be a function on a compact non-empty subset $\mathcal{X}$ of $\mathbb{R}^d$ and $\boldsymbol{x}^\star \in \mathcal{X}$ a maximizer.

**Lipschitz assumption.** We assume that $f$ is Lipschitz with respect to a norm $\|\cdot\|$, that is, there exists $L \geq 0$ such that $\left| f(\boldsymbol{x}) - f(\boldsymbol{y}) \right| \leq L \left\| \boldsymbol{x} - \boldsymbol{y} \right\|$ for all $\boldsymbol{x}, \boldsymbol{y} \in \mathcal{X}$. Furthermore, we assume such a Lipschitz bound $L$ to be known. Even though the smallest Lipschitz constant $\mathrm{Lip}(f) := \min\{L' \geq 0 : f \text{ is } L'\text{-Lipschitz}\}$ is well defined mathematically, it is rarely known exactly in practical black-box problems. As a theoretical curiosity, we will briefly discuss the case $L = \mathrm{Lip}(f)$ (i.e., when the best Lipschitz constant of the unknown black-box function $f$ is known exactly) in Section 4, but for most of our results, we will make the following more realistic assumption.

**Assumption 1.** *For some known Lipschitz constant L, the function $f \colon \mathcal{X} \to \mathbb{R}$ belongs to*
$$\mathcal{F}_L := \left\{ g \colon \mathcal{X} \to \mathbb{R} \mid g \text{ is Lipschitz and } \mathrm{Lip}(g) < L \right\}. \tag{1}$$

The Lipschitzness of $f$ implies the weaker property that $f(\boldsymbol{x}^\star) - f(\boldsymbol{x}) \leq L \left\| \boldsymbol{x}^\star - \boldsymbol{x} \right\|$ for all $\boldsymbol{x} \in \mathcal{X}$, sometimes referred to as *Lipschitzness around a maximizer* $\boldsymbol{x}^\star \in \mathcal{X}$. Although this is not the focus of our work, we will mention when our results hold under this weaker assumption.

**Online learning protocol.** We study the case in which $f$ is black-box, i.e., except for the *a priori* knowledge of $L$, we can only access $f$ by sequentially querying its values at a sequence $\boldsymbol{x}_1, \boldsymbol{x}_2, \ldots \in \mathcal{X}$ of points of our choice. At every round $n \geq 1$, the query point $\boldsymbol{x}_n$ can be chosen as a deterministic function of the values $f(\boldsymbol{x}_1), \ldots, f(\boldsymbol{x}_{n-1})$ observed so far. At the end of round $n$, using all the values $f(\boldsymbol{x}_1), \ldots, f(\boldsymbol{x}_n)$, the learner outputs two quantities:

- a recommendation $\boldsymbol{x}_n^\star \in \mathcal{X}$, with the goal of minimizing the *optimization error* (a.k.a. *simple regret*): $\max(f) - f(\boldsymbol{x}_n^\star)$;
- an *error certificate* $\xi_n \geq 0$, with the constraint to correctly upper bound the optimization error for any $L$-Lipschitz function $f \colon \mathcal{X} \to \mathbb{R}$, i.e., so that $\max(f) - f(\boldsymbol{x}_n^\star) \leq \xi_n$.

We call *certified algorithm* any algorithm for choosing such a sequence $(\boldsymbol{x}_n, \boldsymbol{x}_n^\star, \xi_n)_{n \geq 1}$.

Our goal is to quantify the smallest number of evaluations of $f$ that certified algorithms need in order to find and certify an approximate maximizer of $f$ at accuracy $\varepsilon$. This objective motivates the following definition. For any accuracy $\varepsilon > 0$, we define the *sample complexity* (that could also be called query complexity) of a certified algorithm $A$ for an $L$-Lipschitz function $f$ as
$$\sigma(A, f, \varepsilon) := \inf\left\{ n \geq 1 : \xi_n \leq \varepsilon \right\} \in \{1, 2, \ldots\} \cup \{+\infty\}. \tag{2}$$
This corresponds to the first time when we can stop the algorithm while being sure to have an $\varepsilon$-optimal recommendation $\boldsymbol{x}_n^\star$.

## 1.2 Main Contributions and Outline of the Paper

The main result of this paper is a tight characterization (up to a $\log$ factor) of the optimal sample complexity of certified algorithms in any dimension $d \geq 1$, solving a three-decade old open problem raised by Hansen et al. (1991). More precisely, we prove the following instance-dependent upper and lower bounds, which we later state formally in Theorem 3 of Section 4 (see also discussions therein, as well as Propositions 2 and 3 for the limit case $L = \mathrm{Lip}(f)$).

**Theorem** (Informal statement). *Under a mild geometric assumption on $\mathcal{X}$, there exists a computationally tractable algorithm $A$ (e.g., c.DOO, Algorithm 1) such that, for some constants $C_d, c_d > 0$ (depending exponentially on the dimension d), any Lipschitz function $f \in \mathcal{F}_L$ (see (1)) and any accuracy $\varepsilon$,*
$$\sigma(A, f, \varepsilon) \leq C_d \int_{\mathcal{X}} \frac{\mathrm{d}\boldsymbol{x}}{\left( f(\boldsymbol{x}^\star) - f(\boldsymbol{x}) + \varepsilon \right)^d}, \tag{3}$$
*while any certified algorithm $A'$ must satisfy, for all $f \in \mathcal{F}_L$, and $c \approx c_d(1 - \mathrm{Lip}(f)/L)^d / \log(1/\varepsilon)$,*
$$c \int_{\mathcal{X}} \frac{\mathrm{d}\boldsymbol{x}}{\left( f(\boldsymbol{x}^\star) - f(\boldsymbol{x}) + \varepsilon \right)^d} \leq \sigma(A', f, \varepsilon). \tag{4}$$

In particular, this result extends to any dimension $d \geq 1$ the upper bound proportional to $\int_0^1 \mathrm{d}x / (f(x^\star) - f(x) + \varepsilon)$ that Hansen et al. (1991) derived in dimension $d = 1$ using arguments specific to the geometry of the real line.

**Detailed contributions and outline of the paper.** We make the following contributions.

- As a warmup, we show in Section 2 how to add error certificates to the DOO algorithm (well-known in the more classical zeroth-order Lipschitz optimization setting *without* error certificates, see Perevozchikov 1990; Munos 2011). We then upper bound its sample complexity by the quantity $S_C(f, \varepsilon)$ defined in (5) below. This bound matches a recent bound derived by Bouttier et al. (2020) for a computationally much more expensive algorithm. In passing, we also slightly improve the packing arguments that Munos (2011) used in the non-certified setting.

- In Section 3 we show that, under a mild geometric assumption on $\mathcal{X}$, the complexity measure $S_C(f, \varepsilon)$ is actually proportional to the integral $\int_{\mathcal{X}} \mathrm{d}\boldsymbol{x} / \big(f(\boldsymbol{x}^\star) - f(\boldsymbol{x}) + \varepsilon\big)^d$, which implies (3) above. This extends the bound of Hansen et al. (1991) ($d = 1$) to any dimension $d$.

- Finally, in Section 4, we prove the instance-dependent lower bound (4), which differs from traditional worst-case lower bounds in the Lipschitz setting. Our proof relies on a *local* worst-case analysis that could likely prove useful for other learning tasks.

Some of the proofs are deferred to the Supplementary Material, where we also recall useful results on packing and covering numbers (Section A), as well as provide a slightly improved sample complexity bound on the DOO algorithm in the more classical non-certified setting (Section E).

## 1.3 Related Works

We detail below some connections with the global optimization and the bandit optimization literatures.

**Zeroth-order Lipschitz optimization with error certificates.** The problem of optimizing a function with error certificates has been studied in different settings over the past decades. For instance, in convex optimization, an example of error certificate is given by the *duality gap* between primal and dual feasible points (see, e.g., Boyd and Vandenberghe 2004).

In our setting, namely, global zeroth-order Lipschitz optimization with error certificates, most of the attention seems to have been on the very natural (yet computationally expensive) algorithm introduced by Piyavskii (1972) and Shubert (1972).[1] In dimension $d = 1$, Hansen et al. (1991) proved that its sample complexity $\sigma(\mathrm{PS}, f, \varepsilon)$ for $L$-Lipschitz functions $f : [0, 1] \to \mathbb{R}$ is at most proportional to the integral $\int_0^1 \big(f(x^\star) - f(x) + \varepsilon\big)^{-1} \mathrm{d}x$, and left the question of extending the results to arbitrary dimensions open, stating that the task of "*Extending the results of this paper to the multivariate case appears to be difficult*". Recently, writing $\mathcal{X}_\varepsilon := \{\boldsymbol{x} \in \mathcal{X} : \max(f) - f(\boldsymbol{x}) \leq \varepsilon\}$ for the set of $\varepsilon$-optimal points, $\mathcal{X}_{(a,b]} := \{\boldsymbol{x} \in \mathcal{X} : a < \max(f) - f(\boldsymbol{x}) \leq b\}$ for the set of points in between $a$ and $b$ optimal, and $\mathcal{N}(E, r)$ for the packing number of a set $E$ at scale $r$ (see Section 1.4), Bouttier et al. (2020, Theorem 2) proved a bound valid in any dimension $d \geq 1$ roughly of this form:

$$S_C(f, \varepsilon) := \mathcal{N}\left(\mathcal{X}_\varepsilon, \frac{\varepsilon}{L}\right) + \sum_{k=1}^{m_\varepsilon} \mathcal{N}\left(\mathcal{X}_{(\varepsilon_k, \varepsilon_{k-1}]}, \frac{\varepsilon_k}{L}\right), \tag{5}$$

where the number of terms in the sum is $m_\varepsilon := \big\lceil \log_2(\varepsilon_0 / \varepsilon) \big\rceil$ (with $\varepsilon_0 := L \sup_{\boldsymbol{x}, \boldsymbol{y} \in \mathcal{X}} \|\boldsymbol{x} - \boldsymbol{y}\|$) and the associated scales are given by $\varepsilon_{m_\varepsilon} := \varepsilon$ and $\varepsilon_k := \varepsilon_0 2^{-k}$ for all $k \in \{0, 1, \ldots, m_\varepsilon - 1\}$.

The equivalence we prove in Section 3 between $S_C(f, \varepsilon)$ and $\int_{\mathcal{X}} \mathrm{d}\boldsymbol{x} / (f(\boldsymbol{x}^\star) - f(\boldsymbol{x}) + \varepsilon)^d$ solves in particular the question left open by Hansen et al. (1991). The upper bound we prove for the certified DOO algorithm in Section 2 also indicates that the bound $S_C(f, \varepsilon)$ and the equivalent integral bound can be achieved with a computationally much more tractable algorithm.[2] Indeed, the Piyavskii-Shubert algorithm requires at every step $n$ to solve an inner global Lipschitz optimization problem close to the computation of a Voronoi diagram (see discussion in Bouttier et al. 2020, Section 1.1), hence a running time believed to grow as $n^{\Omega(d)}$ after $n$ function evaluations. On the contrary, as detailed in Remark 1 (Section 2), the running time of our certified version of the DOO algorithm is only of the order of $n \log(n)$ after $n$ of evaluations of $f$.

---

[1]For the interested reader who is unfamiliar with this classic algorithm, we added some details in Section D.3 of the Supplementary Material.

[2]Tractability refers to running time (number of elementary operations) and not number of evaluations of $f$.

**Connections with the bandit optimization literature: upper bounds.** Our work is also strongly connected to the bandit optimization literature, in which multiple authors studied the global Lipschitz optimization problem with zeroth-order (or *bandit*) feedback, either with perfect (deterministic) or noisy (stochastic) observations. In the deterministic setting considered here, these papers show that though the number $(L/\varepsilon)^d$ of evaluations associated to a naive grid search is optimal for worst-case Lipschitz functions (e.g., Thm 1.1.2 by Nesterov 2003), sequential algorithms can approximately optimize more benign functions with a much smaller number of evaluations. Examples of algorithms with such guarantees in the deterministic setting are the branch-and-bound algorithm by Perevozchikov (1990), the DOO algorithm by Munos (2011) or the LIPO algorithm by Malherbe and Vayatis (2017). Examples of algorithms in the stochastic setting are the HOO algorithm by Bubeck et al. (2011) or the (generic yet computationally challenging) Zooming algorithm by Kleinberg et al. (2008, 2019). More examples and references can be found in the textbooks by Munos (2014) and Slivkins (2019).

Note however that, except for the work of Bouttier et al. (2020) mentioned earlier, these bandit optimization papers did not address the problem of *certifying* the accuracy of the recommendations $\boldsymbol{x}_n^\star$. Indeed, all bounds are related to a more classical notion of sample complexity, namely, the minimum number of queries made by an algorithm $A$ before outputting an $\varepsilon$-optimal recommendation:

$$\zeta(A, f, \varepsilon) := \inf\{n \geq 1 : \max(f) - f(\boldsymbol{x}_n^\star) \leq \varepsilon\} \in \{1, 2, \ldots\} \cup \{+\infty\} . \tag{6}$$

Though $\zeta(A, f, \varepsilon)$ is always upper bounded by $\sigma(A, f, \varepsilon)$ defined in (2), these two quantities can differ significantly, as shown by the simple example of constant functions $f$, for which $\zeta(A, f, \varepsilon) = 1$ but $\sigma(A, f, \varepsilon) \approx (L/\varepsilon)^d$ since the only way to *certify* that the output is $\varepsilon$-optimal is essentially to perform a grid-search with step-size roughly $\varepsilon/L$, so as to be sure there is no hidden bump of height more than $\varepsilon$. At a high level, the more "constant" a function is, the easier it is to recommend an $\varepsilon$-optimal point, but the harder it is to certify that such recommendation is actually a good recommendation. See the Supplementary Material (Section E) for a comparison of bounds.

Despite this important difference, the bandit viewpoint (using packing numbers instead of more specific one-dimensional arguments) is key to obtain our multi-dimensional integral characterization.

**Comparison with existing lower bounds.** Several lower bounds were derived in the bandit Lipschitz optimization setting without error certificates. When rewritten in terms of the accuracy $\varepsilon$ and translated into our deterministic setting, the lower bounds of Horn (2006) (when $d^\star = d/2$) and of Bubeck et al. (2011) (for any $d^\star$) are of the form $\inf_A \sup_{f \in \mathcal{G}_{d^\star}} \zeta(A, f, \varepsilon) \gtrsim (1/\varepsilon)^{d^\star}$, where $\mathcal{G}_{d^\star}$ is the subset of $L$-Lipschitz functions with near-optimality dimension at most $d^\star$. These are worst-case (minimax) lower bounds.

On the contrary, our instance-dependent lower bound (4) quantifies the minimum number of evaluations to certify an $\varepsilon$-optimal point *for each function* $f \in \mathcal{F}_L$. Note that here the certified setting enables to obtain meaningful instance-dependent lower bounds, whereas their non-certified counterparts would be trivial (equal to one). Our proof relies on a *local* worst-case analysis in the same spirit as for distribution-dependent lower bounds in stochastic multi-armed bandits (see, e.g., Theorem 16.2 in Lattimore and Szepesvári 2020), yet for continuous instead of finite action sets. We believe this lower bound technique should prove useful for other learning tasks.

## 1.4 Recurring Notation

This short section contains a summary of all the notation that we use in the paper and can be used by the reader for easy referencing. We denote the set of positive integers $\{1, 2, \ldots\}$ by $\mathbb{N}^*$ and let $\mathbb{N} := \mathbb{N}^* \cup \{0\}$. For all $n \in \mathbb{N}^*$, we denote by $[n]$ the set of the first $n$ integers $\{1, \ldots, n\}$. We denote the Lebesgue measure of a (Lebesgue-measurable) set $E$ by $\mathrm{vol}(E)$ and refer to it simply as its *volume*. For all $\rho > 0$ and $\boldsymbol{x} \in \mathbb{R}^d$, we denote by $B_\rho(\boldsymbol{x})$ the closed ball with radius $\rho$ centered at $\boldsymbol{x}$, with respect to the arbitrary norm $\|\cdot\|$ that is fixed throughout the paper. We also write $B_\rho$ for the ball with radius $\rho$ centered at the origin and denote by $v_\rho$ its volume.

$\mathrm{Lip}(f)$ denotes the smallest Lipschitz constant of our target $L$-Lipschitz function $f : \mathcal{X} \to \mathbb{R}$. The set of its $\varepsilon$-optimal points is denoted by $\mathcal{X}_\varepsilon := \{\boldsymbol{x} \in \mathcal{X} : \max(f) - f(\boldsymbol{x}) \leq \varepsilon\}$, its complement (i.e., the set of $\varepsilon$-*suboptimal points*) by $\mathcal{X}_\varepsilon^c$, and for all $0 \leq a < b$, the $(a, b]$-*layer* (i.e., the set of points that are $b$-optimal but $a$-suboptimal) by $\mathcal{X}_{(a,b]} := \mathcal{X}_a^c \cap \mathcal{X}_b = \{\boldsymbol{x} \in \mathcal{X} : a < f(\boldsymbol{x}^*) - f(\boldsymbol{x}) \leq b\}$. Since $f$ is $L$-Lipschitz, every point in $\mathcal{X}$ is $\varepsilon_0$-optimal, with $\varepsilon_0$ defined by $\varepsilon_0 := L \max_{\boldsymbol{x}, \boldsymbol{y} \in \mathcal{X}} \|\boldsymbol{x} - \boldsymbol{y}\|$. In

other words, $\mathcal{X}_{\varepsilon_0} = \mathcal{X}$. For this reason, without loss of generality, we will only consider values of accuracy $\varepsilon$ smaller than or equal to $\varepsilon_0$.

For any bounded set $E \subset \mathbb{R}^d$ and all $r > 0$, the $r$-*packing number* of $E$ is the largest cardinality of an $r$-packing of $E$, that is, $\mathcal{N}(E, r) := \sup\{k \in \mathbb{N}^* : \exists \boldsymbol{x}_1, \ldots, \boldsymbol{x}_k \in E, \min_{i \neq j} \|\boldsymbol{x}_i - \boldsymbol{x}_j\| > r\}$ if $E$ is nonempty, zero otherwise; the $r$-*covering number* of $E$ is the smallest cardinality of an $r$-covering of $E$, i.e., $\mathcal{M}(E, r) := \min\{k \in \mathbb{N}^* : \exists \boldsymbol{x}_1, \ldots, \boldsymbol{x}_k \in \mathbb{R}^d, \forall \boldsymbol{x} \in E, \exists i \in [k], \|\boldsymbol{x} - \boldsymbol{x}_i\| \leq r\}$ if $E$ is nonempty, zero otherwise. Well-known and useful properties of packing (and covering) numbers are recalled in Section A of the Supplementary Material.

## 2 Warmup: Certified DOO Has Sample Complexity $S_{\mathrm{C}}(f, \varepsilon)$

In this section, we start by adapting the well-known DOO algorithm (Perevozchikov, 1990; Munos, 2011) to the certified setting. We then bound its sample complexity by the quantity $S_{\mathrm{C}}(f, \varepsilon)$ defined in Eq. (5). In passing, we slightly improve the packing arguments used by Munos (2011) in the non-certified setting (Supplementary Material, Section E). In Section 4, we will prove that this bound is optimal (up to logarithmic factors) for certified algorithms.

The certified DOO algorithm (c.DOO, Algorithm 1) is defined for a fixed $K \in \mathbb{N}^*$, by an infinite sequence of subsets of $\mathcal{X}$ of the form $(X_{h,i})_{h \in \mathbb{N}, i=0, \ldots, K^h - 1}$, called *cells*. For each $h \in \mathbb{N}$, the cells $X_{h,0}, \ldots, X_{h, K^h - 1}$ are non-empty, pairwise disjoint, and their union contains $\mathcal{X}$. The sequence $(X_{h,i})_{h \in \mathbb{N}, i=0, \ldots, K^h - 1}$ is associated with a $K$-ary tree in the following way. For any $h \in \mathbb{N}$ and $j \in \{0, \ldots, K^h - 1\}$, there exist $K$ distinct $i_1, \ldots, i_K \in \{0, \ldots, K^{h+1} - 1\}$ such that $X_{h+1, i_1}, \ldots, X_{h+1, i_K}$ form a partition of $X_{h,j}$. We call $(h+1, i_1), \ldots, (h+1, i_K)$ the *children* of $(h, j)$. To each cell $X_{h,i}$ ($h \in \mathbb{N}$, $i \in \{0, \ldots, K^h - 1\}$) is associated a *representative* $\boldsymbol{x}_{h,i} \in X_{h,i}$, which can be thought of as the "center" of the cell. We assume that feasible cells have feasible representatives, i.e., that $X_{h,i} \cap \mathcal{X} \neq \varnothing$ implies $\boldsymbol{x}_{h,i} \in \mathcal{X}$. The two following assumptions prescribe a sufficiently good behavior of the sequences of cells and representatives.

**Assumption 2.** *There exist two positive constants $\delta \in (0, 1)$ and $R > 0$ such that, for any cell $X_{h,i}$ ($h \in \mathbb{N}$, $i = 0, \ldots, K^h - 1$) and all $\boldsymbol{u}, \boldsymbol{v} \in X_{h,i}$, it holds that $\|\boldsymbol{u} - \boldsymbol{v}\| \leq R\delta^h$.*

**Assumption 3.** *There exists $\nu > 0$ such that, with $\delta$ as in Assumption 2, for any $h \in \mathbb{N}$, $i = 0, \ldots, K^h - 1$, $h' \in \mathbb{N}$, $i' = 0, \ldots, K^{h'} - 1$, with $(h, i) \neq (h', i')$, $\|\boldsymbol{x}_{h,i} - \boldsymbol{x}_{h',i'}\| \geq \nu\delta^{\max(h, h')}$.*

The classic Assumption 2 is simply stating that diameters of cells decrease geometrically with the depth of the tree. Assumption 3, which is key for our improved analysis, is slightly stronger than the corresponding one in Munos (2011), yet very easy to satisfy. Indeed, one can prove that for any compact $\mathcal{X}$, it is always possible to find a sequence of cells and representatives satisfying Assumptions 2 and 3. For instance, if $\mathcal{X}$ is the unit hypercube $[0, 1]^d$ and $\|\cdot\|$ is the supremum norm $\|\cdot\|_\infty$, we can define cells by bisection, letting $K = 2^d$, $X_{h,i}$ be a hypercube of edge-length $2^{-h}$, and $\boldsymbol{x}_{h,i}$ be its center (for $h \in \mathbb{N}$ and $i = 0, \ldots, 2^{dh} - 1$). In this case, Assumptions 2 and 3 are satisfied with $R = 1$ and $\delta = \nu = 1/2$.

Our certified version of the DOO algorithm (c.DOO, Algorithm 1) maintains a set of indices of *active* cells $\mathcal{L}_n$ throughout rounds $n$. During each iteration $k$, it selects the index of the most promising active cell $(h^\star, i^\star)$ (Line 5) and splits it into its $K$ children $\mathcal{L}_+$ (Line 6). Then, it sequentially picks the representatives of the cells corresponding to each of these children (Line 10), observes the value of the target function $f$ at these points (Line 11), recommends the point with the highest observed value of $f$ (Line 12), and outputs a certificate $\xi_n = \big(f(\boldsymbol{x}_{h^\star, i^\star}) + LR\delta^{h^\star}\big) - f(\boldsymbol{x}_n^\star)$ (Line 13) that is the difference between an upper bound on $\max(f)$ and the currently recommended value $f(\boldsymbol{x}_n^\star)$. In the meantime, all children in $\mathcal{L}_+$ are added to the set of active indices $\mathcal{L}_n$ (Line 9), and the current iteration is concluded by removing $(h^\star, i^\star)$ from $\mathcal{L}_n$ (Line 14), now that it has been replaced by its refinement $\mathcal{L}_+$.

**Remark 1.** *The running-time of c.DOO (ignoring the cost of calling the function $f$) is driven by the computation of the recommendation $\boldsymbol{x}_n^\star \in \operatorname{argmax}_{\boldsymbol{x} \in \{\boldsymbol{x}_1, \ldots, \boldsymbol{x}_n\}} f(\boldsymbol{x})$ (Line 12) and the search of the index of the most promising active cell $(h^\star, i^\star) \in \operatorname{argmax}_{(h,i) \in \mathcal{L}_n} \{f(\boldsymbol{x}_{h,i}) + LR\delta^h\}$ (Line 5). The recommendation $\boldsymbol{x}_n^\star$ can be computed sequentially in constant time (by comparing the new value $f(\boldsymbol{x}_n)$ with the current maximum). In Line 5, the leaf $(h^\star, i^\star)$ to be split at iteration $k$ can be*

---

**Algorithm 1:** Certified DOO (c.DOO)

---

**input:** $\mathcal{X}, L, K, \delta, R$, cells $(X_{h,i})_{h\in\mathbb{N},i=0,\ldots,K^h-1}$, representatives $(\boldsymbol{x}_{h,i})_{h\in\mathbb{N},i=0,\ldots,K^h-1}$
**initialization:** let $n \leftarrow 1$ and $\mathcal{L}_1 \leftarrow \{(0,0)\}$

1   pick the first query point $\boldsymbol{x}_1 \leftarrow \boldsymbol{x}_{0,0}$
2   observe the value $f(\boldsymbol{x}_1)$
3   output recommendation $\boldsymbol{x}_1^\star \leftarrow \boldsymbol{x}_1$ and error certificate $\xi_1 \leftarrow LR$
4   **for** *iteration* $k = 1, 2, \ldots$ **do**
5      let $(h^\star, i^\star) \in \mathrm{argmax}_{(h,i)\in\mathcal{L}_n}\big\{f(\boldsymbol{x}_{h,i}) + LR\delta^h\big\}$    `// ties broken arbitrarily`
6      let $\mathcal{L}_+$ be the set of the $K$ children of $(h^\star, i^\star)$
7      **for each** *child* $(h^\star + 1, j) \in \mathcal{L}_+$ *of* $(h^\star, i^\star)$ **do**
8          **if** $X_{h^\star+1,j} \cap \mathcal{X} \neq \varnothing$ **then**
9              let $n \leftarrow n+1$ and $\mathcal{L}_n \leftarrow \mathcal{L}_{n-1} \cup \{(h^\star+1, j)\}$
10            pick the next query point $\boldsymbol{x}_n \leftarrow \boldsymbol{x}_{h^\star+1,j}$
11            observe the value $f(\boldsymbol{x}_n)$
12            output a recommendation $\boldsymbol{x}_n^\star \in \mathrm{argmax}_{\boldsymbol{x}\in\{\boldsymbol{x}_1,\ldots,\boldsymbol{x}_n\}} f(\boldsymbol{x})$
13            output the error certificate $\xi_n \leftarrow f(\boldsymbol{x}_{h^\star,i^\star}) + LR\delta^{h^\star} - f(\boldsymbol{x}_n^\star)$
14      remove $(h^\star, i^\star)$ from $\mathcal{L}_n$

---

*computed sequentially in logarithmic time (using a max-heap structure). Therefore, the running time of* c.DOO *is of order* $n\log(n)$ *in the number* $n$ *of evaluations of the function* $f$.

The next proposition shows that the sample complexity (see (2)) of the certified DOO algorithm is upper bounded (up to constants) by the instance-dependent quantity $S_{\mathrm{C}}(f,\varepsilon)$ introduced in Eq. (5).

**Proposition 1.** *If Assumptions 2 and 3 hold, then Algorithm 1 is a certified algorithm and letting* $a_d := 2 + K\big(\mathbf{1}_{\nu/R\geq 1} + \mathbf{1}_{\nu/R<1}(4R/\nu)^d\big)$, *its sample complexity satisfies, for all Lipschitz functions* $f \in \mathcal{F}_L$ *(see (1)) and any accuracy* $\varepsilon \in (0, \varepsilon_0]$,

$$\sigma(\mathrm{c.DOO}, f, \varepsilon) \leq a_d S_{\mathrm{C}}(f, \varepsilon) .$$

The proof is postponed to Section B of the Supplementary Material and shares some arguments with those of Perevozchikov (1990) and Munos (2011), originally written for the non-certified setting. The key change is to partition the values of $f$, instead of its domain $\mathcal{X}$ at any depth $h$ of the tree (see Munos 2011), when counting the representatives selected at all levels. The idea of using layers $\mathcal{X}_{(\varepsilon_i,\,\varepsilon_{i-1}]}$ was already present in Kleinberg et al. (2008, 2019) and Bouttier et al. (2020) for more computationally challenging algorithms (see discussion in Section 1.3).

## 3   Characterization of $S_{\mathrm{C}}(f, \varepsilon)$

Earlier, we mentioned that the quantity $S_{\mathrm{C}}(f,\varepsilon)$ introduced in Eq. (5) upper bounds the sample complexity of several certified algorithms, such as c.DOO or Piyavskii-Shubert. In this section, we provide a characterization of this quantity in terms of a much cleaner and integral expression.

This result is inspired by Hansen et al. (1991), that in dimension $d = 1$, derive an elegant bound on the sample complexity $\sigma(\mathrm{PS}, f, \varepsilon)$ of the certified Piyavskii-Shubert algorithm for any Lipschitz function $f$ and accuracy $\varepsilon$. They proved that $\sigma(\mathrm{PS}, f, \varepsilon)$ is upper bounded by $\int_0^1 \mathrm{d}x/(f(x^\star) - f(x) + \varepsilon)$ up to constants. However, the authors rely heavily on the one-dimensional assumption and the specific form of the Piyavskii-Shubert algorithm in this setting, stating that the task of "*Extending the results of this paper to the multivariate case appears to be difficult*". In this section, we show an equivalence between $S_{\mathrm{C}}(f,\varepsilon)$ and this type of integral bound in any dimension $d$. Putting this together with a recent result of Bouttier et al. (2020) (which proves that up to constants, $\sigma(\mathrm{PS}, f, \varepsilon) \leq S_{\mathrm{C}}(f,\varepsilon)$) solves the long-standing problem raised by Hansen et al. (1991) three decades ago.

To tame the wild spectrum of shapes that compact subsets may have, we will assume that $\mathcal{X}$ satisfies the following mild geometric assumption. At a high level, it says that a constant fraction of each (sufficiently small) ball centered at a point in $\mathcal{X}$ is included in $\mathcal{X}$. This removes sets containing isolated points or cuspidal corners and, as can be shown quickly, includes most domains that are typically used, such as finite unions of convex sets with non-empty interiors. This natural assumption

is weaker than the classic rolling ball assumption from the statistics literature (Cuevas et al., 2012; Walther, 1997) and has already been proved useful in the past (Hu et al., 2020).

**Assumption 4.** *There exist two constants $r_0 > 0, \gamma \in (0,1]$ such that, for any $\boldsymbol{x} \in \mathcal{X}$ and all $r \in (0, r_0)$, $\mathrm{vol}\big(B_r(\boldsymbol{x}) \cap \mathcal{X}\big) \geq \gamma v_r$.*

Note that, above, $\gamma$ may implicitly depend on $d$. For instance if $\mathcal{X}$ is an hypercube, then $\gamma$ is at most $2^{-d}$. We can now state the main result of this section. Its proof relies on some additional technical results that are deferred to the Supplementary Material. Recall that $S_{\mathrm{C}}(f, \varepsilon)$ was defined in (5).

**Theorem 1.** *For any Lipschitz function $f \in \mathcal{F}_L$ (see (1)), if Assumption 4 holds with $r_0 > \varepsilon_0/2L, \gamma \in (0,1]$,[3] then there exist $c_d, C_d > 0$ (e.g., $c_d := 1/v_{1/L}$ and $C_d := 1/(\gamma v_{1/128L})$) such that, for all $\varepsilon \in (0, \varepsilon_0]$,*

$$c_d \int_{\mathcal{X}} \frac{\mathrm{d}\boldsymbol{x}}{\big(f(\boldsymbol{x}^\star) - f(\boldsymbol{x}) + \varepsilon\big)^d} \leq S_{\mathrm{C}}(f, \varepsilon) \leq C_d \int_{\mathcal{X}} \frac{\mathrm{d}\boldsymbol{x}}{\big(f(\boldsymbol{x}^\star) - f(\boldsymbol{x}) + \varepsilon\big)^d} \ .$$

Remark that in Theorem 1, the integral is multiplied by $L^d$ in the two inequalities. This is to compensate for the fact that this integral does not depend on $L$, while $S_{\mathrm{C}}(f, \varepsilon)$ scales like $L^d$ for a fixed $f$ as $L \to \infty$.

*Proof.* Fix any $\varepsilon \in (0, \varepsilon_0]$ and recall that $m_\varepsilon := \big\lceil \log_2(\varepsilon_0/\varepsilon) \big\rceil$, $\varepsilon_{m_\varepsilon} := \varepsilon$, and for all $k \leq m_\varepsilon - 1$, $\varepsilon_k := \varepsilon_0 2^{-k}$. Partition the domain of integration $\mathcal{X}$ into the following $m_\varepsilon + 1$ sets: the set of $\varepsilon$-optimal points $\mathcal{X}_\varepsilon$ and the $m_\varepsilon$ layers $\mathcal{X}_{(\varepsilon_k, \varepsilon_{k-1}]}$, for $k \in [m_\varepsilon]$. We begin by proving the first inequality:

$$\int_{\mathcal{X}} \frac{\mathrm{d}\boldsymbol{x}}{\big(f(\boldsymbol{x}^\star) - f(\boldsymbol{x}) + \varepsilon\big)^d} \leq \frac{\mathrm{vol}(\mathcal{X}_\varepsilon)}{\varepsilon^d} + \sum_{k=1}^{m_\varepsilon} \frac{\mathrm{vol}\big(\mathcal{X}_{(\varepsilon_k, \varepsilon_{k-1}]}\big)}{(\varepsilon_k + \varepsilon)^d}$$

$$\leq \frac{\mathcal{M}\big(\mathcal{X}_\varepsilon, \frac{\varepsilon}{L}\big) \cdot v_1\big(\frac{\varepsilon}{L}\big)^d}{\varepsilon^d} + \sum_{k=1}^{m_\varepsilon} \frac{\mathcal{M}\big(\mathcal{X}_{(\varepsilon_k, \varepsilon_{k-1}]}, \frac{\varepsilon_k}{L}\big) \cdot v_1\big(\frac{\varepsilon_k}{L}\big)^d}{\varepsilon_k^d}$$

$$\leq \frac{v_1}{L^d} \left( \mathcal{N}\left(\mathcal{X}_\varepsilon, \frac{\varepsilon}{L}\right) + \sum_{k=1}^{m_\varepsilon} \mathcal{N}\left(\mathcal{X}_{(\varepsilon_k, \varepsilon_{k-1}]}, \frac{\varepsilon_k}{L}\right) \right) \ ,$$

where the first inequality follows by lower bounding $f(\boldsymbol{x}^\star) - f$ with its infimum on the partition, the second one by dropping $\varepsilon > 0$ from the second denominator and upper bounding the volume of a set with the volume of the balls of a smallest $\varepsilon_k/L$-cover, and the last one by the fact that covering numbers are always smaller than packing numbers (we recall this known result in the Supplementary Material, Lemma 1, Eq. (8)). This proves the first part of the theorem.

For the second one, we have

$$\int_{\mathcal{X}} \frac{\mathrm{d}\boldsymbol{x}}{\big(f(\boldsymbol{x}^\star) - f(\boldsymbol{x}) + \varepsilon\big)^d} \geq \frac{\mathrm{vol}(\mathcal{X}_\varepsilon)}{(\varepsilon + \varepsilon)^d} + \sum_{k=1}^{m_\varepsilon} \frac{\mathrm{vol}\big(\mathcal{X}_{(\varepsilon_k, \varepsilon_{k-1}]}\big)}{(\varepsilon_{k-1} + \varepsilon)^d}$$

$$\geq \frac{1}{2^d} \frac{\mathrm{vol}(\mathcal{X}_\varepsilon)}{\varepsilon^d} + \frac{1}{4^d} \sum_{k=1}^{m_\varepsilon} \frac{\mathrm{vol}\big(\mathcal{X}_{(\varepsilon_k, \varepsilon_{k-1}]}\big)}{\varepsilon_k^d} \geq \frac{1}{32^d} \left( \frac{\mathrm{vol}(\mathcal{X}_{2\varepsilon})}{\varepsilon^d} + \sum_{k=1}^{m_\varepsilon} \frac{\mathrm{vol}\big(\mathcal{X}_{(\frac{1}{2}\varepsilon_k, 2\varepsilon_{k-1}]}\big)}{\varepsilon_{k-1}^d} \right)$$

$$\geq \frac{\mathcal{N}\big(\mathcal{X}_\varepsilon, \frac{\varepsilon}{L}\big) \mathrm{vol}\big(\frac{\varepsilon}{2L}B_1\big)}{(32\,\varepsilon)^d/\gamma} + \sum_{k=1}^{m_\varepsilon} \frac{\mathcal{N}\big(\mathcal{X}_{(\varepsilon_k, \varepsilon_{k-1}]}, \frac{\varepsilon_k}{L}\big) \mathrm{vol}\big(\frac{\varepsilon_k}{2L}B_1\big)}{(32\,\varepsilon_{k-1})^d/\gamma}$$

$$\geq \gamma v_{1/64L} \mathcal{N}\left(\mathcal{X}_\varepsilon, \frac{\varepsilon}{L}\right) + \gamma v_{1/128L} \sum_{k=1}^{m_\varepsilon} \mathcal{N}\left(\mathcal{X}_{(\varepsilon_k, \varepsilon_{k-1}]}, \frac{\varepsilon_k}{L}\right) \ ,$$

where the first inequality follows by upper bounding $f(\boldsymbol{x}^\star) - f$ with its supremum on the partition, the second one by $\varepsilon \leq \varepsilon_{k-1}$ (for all $k \in [m_\varepsilon + 1]$) and $\varepsilon_{k-1} \leq 2\varepsilon_k$ (for all $k \in [m_\varepsilon]$), the third

---

[3]We actually prove a stronger result. The first inequality holds more generally for any $f$ that is $L$-Lipschitz around a maximizer and Lebesgue-measurable, and does not require $\mathcal{X}$ to satisfy Assumption 4.

one by lower bounding the sum of disjoint layers with that of overlapping ones (proved in the Supplementary Material, Lemma 3), and the fourth one by the elementary inclusions $\mathcal{X}_{2\varepsilon} \supseteq \mathcal{X}_{\frac{3}{2}\varepsilon}$ and $\mathcal{X}_{\left(\frac{1}{2}\varepsilon_k,\, 2\varepsilon_{k-1}\right]} \supseteq \mathcal{X}_{\left(\frac{1}{2}\varepsilon_k,\, \frac{3}{2}\varepsilon_{k-1}\right]}$ (for all $k \in [m_\varepsilon]$) followed by a relationship between packing numbers and volumes (proved in the Supplementary Material, Proposition 4) that holds under Assumption 4. $\qquad\square$

## 4  Optimality of $S_{\mathrm{C}}(f, \varepsilon)$

We begin this section by proving an $f$-dependent lower bound on the sample complexity of certified algorithms that matches the upper bound $S_{\mathrm{C}}(f, \varepsilon)$ on the sample complexity of the c.DOO algorithm (Proposition 1), up to a $\log(1/\varepsilon)$ term, a dimension-dependent constant, and the term $(1 - \mathrm{Lip}(f)/L)^d$.

The proof of this result differs from those of traditional worst-case lower bounds. The idea is to build a *local* worst-case analysis, introducing a weaker notion of sample complexity $\tau$ that is smaller than $\sigma(A, f, \varepsilon)$. Then we further lower bound this quantity by finding adversarial perturbations of the target function $f$ with the following opposing properties. First, these perturbations have to be similar enough to $f$ so that running $A$ on them would return the same recommendations for sufficiently many rounds. Second, they have to be different enough from $f$ so that enough rounds have to pass before being able to certify sub-$\varepsilon$ accuracy. We recall that $m_\varepsilon := \left\lceil \log_2(\varepsilon_0/\varepsilon) \right\rceil$ and that $S_{\mathrm{C}}(f, \varepsilon)$ was defined in (5).

**Theorem 2.** *The sample complexity of any certified algorithm $A$ satisfies*

$$\sigma(A, f, \varepsilon) > \frac{c'_d (1 - \mathrm{Lip}(f)/L)^d}{1 + m_\varepsilon} S_{\mathrm{C}}(f, \varepsilon)$$

*for some constant $c'_d$ (that can be taken as $c'_d = 2^{-2}2^{-7d}$), any Lipschitz function $f \in \mathcal{F}_L$ (see (1)) and all $\varepsilon \in (0, \varepsilon_0]$.*

*Proof.* Fix any $f \in \mathcal{F}_L$ and an accuracy $\varepsilon \in (0, \varepsilon_0]$. We begin by defining the tightest error certificate that a certified algorithm $A$ could return based on its first $n$ observations of $f$. Formally,

$$\mathrm{err}_n(A) := \sup\big\{\max(g) - g(\boldsymbol{x}_n^\star) : g \text{ is } L\text{-Lipschitz and } f(\boldsymbol{x}) = g(\boldsymbol{x}) \text{ for all } x \in \{\boldsymbol{x}_1, \ldots, \boldsymbol{x}_n\}\big\}$$

where $\boldsymbol{x}_i = \boldsymbol{x}_i(A, f)$ and $\boldsymbol{x}_i^\star = \boldsymbol{x}_i^\star(A, f)$ are the query and recommendation points chosen by $A$ at time $i$ when run on $f$ (to lighten the notation, we omit the explicit dependencies on $A$ and $f$ of $\boldsymbol{x}_i$ and $\boldsymbol{x}_i^\star$). In particular, $\max(f) - f(\boldsymbol{x}_n^\star) \leq \mathrm{err}_n(A)$. Based on this quantity, we define a corresponding notion of optimal sample complexity $\tau$ as the smallest number of rounds $n$ that the best certified algorithm $A'$ needs in order to guarantee that $\mathrm{err}_n(A') \leq \varepsilon$. Formally,

$$\tau := \min\big\{n \in \mathbb{N}^* : \inf_{A'}\big(\mathrm{err}_n(A')\big) \leq \varepsilon\big\}, \tag{7}$$

where the infimum is over all certified algorithms $A'$. It is immediate to prove that $\tau$ is finite, by considering an algorithm that queries a dense sequence of points (independently of the observed function values) and outputs a recommendation that maximizes the observed values.

Crucially, $\tau$ lower bounds the sample complexity $\sigma(A, f, \varepsilon)$ of any certified algorithm $A$. At a high level, this makes intuitive sense because $\tau$ guarantees a weaker property: that the best certificate of the best algorithm is small, while $\sigma(A, f, \varepsilon)$ controls the certificate of the specific algorithm $A$. We defer a formal proof of this statement to the Supplementary Material, Section D.1.

Since $\sigma(A, f, \varepsilon) \geq \tau$, to prove the theorem it is sufficient to show that, with $c := 2^{-2}2^{-7d}(1 - \mathrm{Lip}(f)/L)^d$,

$$\tau > \frac{c}{1 + m_\varepsilon} S_{\mathrm{C}}(f, \varepsilon).$$

Let $Q := 16L/\big(L - \mathrm{Lip}(f)\big)$. If $S_{\mathrm{C}}(f, \varepsilon)/(1 + m_\varepsilon) < 3(8Q)^d$, then $\tau \geq 1 > 3/4 > cS_{\mathrm{C}}(f, \varepsilon)/(1 + m_\varepsilon)$. Assume from now on that $S_{\mathrm{C}}(f, \varepsilon)/(1 + m_\varepsilon) \geq 3(8Q)^d$.

The idea now is to upper bound the sum of $1 + m_\varepsilon$ packing numbers that define $S_{\mathrm{C}}(f, \varepsilon)$ in (5) with the largest one multiplied by $1 + m_\varepsilon$. This way, $S_{\mathrm{C}}(f, \varepsilon)/(1 + m_\varepsilon)$ can be controlled by (an upper bound of) the largest summand in $S_{\mathrm{C}}(f, \varepsilon)$. Let $\tilde{\varepsilon}$ be the scale achieving the maximum in (5), that is

$$\tilde{\varepsilon} = \begin{cases} \varepsilon, & \text{if } \mathcal{N}\left(\mathcal{X}_\varepsilon, \frac{\varepsilon}{L}\right) \geq \max_{i \in \{1, \ldots, m_\varepsilon\}} \mathcal{N}\left(\mathcal{X}_{(\varepsilon_i,\, \varepsilon_{i-1}]}, \frac{\varepsilon_i}{L}\right), \\ \varepsilon_{i^\star - 1}, & \text{otherwise, where } i^\star \in \mathrm{argmax}_{i \in \{1, \ldots, m_\varepsilon\}} \mathcal{N}\left(\mathcal{X}_{(\varepsilon_i,\, \varepsilon_{i-1}]}, \frac{\varepsilon_i}{L}\right). \end{cases}$$

Since $\mathcal{N}\left(\mathcal{X}_\varepsilon,\ \varepsilon/L\right) \leq \mathcal{N}\left(\mathcal{X}_\varepsilon,\ \varepsilon/2L\right)$ and $\mathcal{N}\left(\mathcal{X}_{(\varepsilon_i,\ \varepsilon_{i-1}]},\ \varepsilon_i/L\right) \leq \mathcal{N}\left(\mathcal{X}_{\varepsilon_{i-1}},\ \varepsilon_{i-1}/2L\right)$, we then have $S_{\mathrm{C}}(f,\varepsilon) \leq (m_\varepsilon + 1)\mathcal{N}\left(\mathcal{X}_{\tilde{\varepsilon}},\ \tilde{\varepsilon}/2L\right)$. Let now $n \leq cS_{\mathrm{C}}(f,\varepsilon)/(1+m_\varepsilon)$. We then have $n \leq c\mathcal{N}\left(\mathcal{X}_{\tilde{\varepsilon}},\ \tilde{\varepsilon}/2L\right)$. From a known property of packing numbers (Supplementary Material, Lemma 2),

$$\mathcal{N}\left(\mathcal{X}_{\tilde{\varepsilon}},\ \frac{Q\tilde{\varepsilon}}{L}\right) \geq \left(\frac{1}{8Q}\right)^d \mathcal{N}\left(\mathcal{X}_{\tilde{\varepsilon}},\ \frac{\tilde{\varepsilon}}{2L}\right) \geq \left(\frac{1}{8Q}\right)^d \frac{S_{\mathrm{C}}(f,\varepsilon)}{m_\varepsilon + 1} \geq 3\ ,$$

by our initial assumption. Then we have $n \leq c(8Q)^d \mathcal{N}\left(\mathcal{X}_{\tilde{\varepsilon}},\ Q\tilde{\varepsilon}/L\right)$. Since $c(8Q)^d = 1/4$, we thus obtain $n \leq \mathcal{N}\left(\mathcal{X}_{\tilde{\varepsilon}},\ Q\tilde{\varepsilon}/L\right) - 2$.

Consider a certified algorithm $A$ for $L$-Lipschitz functions. Fix a $(Q\tilde{\varepsilon}/L)$-packing $\tilde{\boldsymbol{x}}_1,\ldots,\tilde{\boldsymbol{x}}_N$ of $\mathcal{X}_{\tilde{\varepsilon}}$ with cardinality $N = \mathcal{N}\left(\mathcal{X}_{\tilde{\varepsilon}},\ Q\tilde{\varepsilon}/L\right)$. Then the open balls of centers $\tilde{\boldsymbol{x}}_1,\ldots,\tilde{\boldsymbol{x}}_N$ and radius $Q\tilde{\varepsilon}/2L$ are disjoint and two of them, with centers, say, $\tilde{\boldsymbol{x}}_1$ and $\tilde{\boldsymbol{x}}_2$, do not contain any of the points $\boldsymbol{x}_1,\ldots,\boldsymbol{x}_n$ queried by $A$ when it is run on $f$. Let, for $\boldsymbol{x} \in \mathcal{X}$,

$$g_{\tilde{\varepsilon}}(\boldsymbol{x}) := \left(8\tilde{\varepsilon} - \frac{16L}{Q}\left\|\boldsymbol{x} - \tilde{\boldsymbol{x}}_1\right\|\right)\mathbb{I}\left\{\boldsymbol{x} \in \mathcal{X} \cap B_{Q\tilde{\varepsilon}/2L}(\tilde{\boldsymbol{x}}_1)\right\}\ .$$

Then $g_{\tilde{\varepsilon}}(x)$ is $16L/Q = L - \mathrm{Lip}(f)$ Lipschitz. Hence $f + g_{\tilde{\varepsilon}}$ and $f - g_{\tilde{\varepsilon}}$ are $L$-Lipschitz. Note that $f$, $f + g_{\tilde{\varepsilon}}$ and $f - g_{\tilde{\varepsilon}}$ coincide on the points $\boldsymbol{x}_1,\ldots,\boldsymbol{x}_n$ that $A$ queries when it is run on $f$. As a consequence, $A$ queries the same points and returns the same recommendation $\boldsymbol{x}_n^\star$ when it is run on any of the three functions $f$, $f + g_{\tilde{\varepsilon}}$, and $f - g_{\tilde{\varepsilon}}$.

Consider first the case $\boldsymbol{x}_n^\star \in B(\tilde{\boldsymbol{x}}_1, Q\tilde{\varepsilon}/4L)$. Then, we have, by definition of $g_{\tilde{\varepsilon}}$ and the fact that $\tilde{\boldsymbol{x}}_2 \in \mathcal{X}_{\tilde{\varepsilon}}$, $f(\tilde{\boldsymbol{x}}_2) - g_{\tilde{\varepsilon}}(\tilde{\boldsymbol{x}}_2) - f(\boldsymbol{x}_n^\star) + g_{\tilde{\varepsilon}}(\boldsymbol{x}_n^\star) \geq -\tilde{\varepsilon} + 8\tilde{\varepsilon} - \frac{16L}{Q}\frac{Q\tilde{\varepsilon}}{4L} = 3\tilde{\varepsilon}$.

Now consider the case $\boldsymbol{x}_n^\star \notin B(\tilde{\boldsymbol{x}}_1, Q\tilde{\varepsilon}/4L)$. Then, we have, by definition of $g_{\tilde{\varepsilon}}$ and the fact that $\tilde{\boldsymbol{x}}_1 \in \mathcal{X}_{\tilde{\varepsilon}}$, $f(\tilde{\boldsymbol{x}}_1) + g_{\tilde{\varepsilon}}(\tilde{\boldsymbol{x}}_1) - f(\boldsymbol{x}_n^\star) - g_{\tilde{\varepsilon}}(\boldsymbol{x}_n^\star) \geq -\tilde{\varepsilon} + 8\tilde{\varepsilon} - 8\tilde{\varepsilon} + \frac{16L}{Q}\frac{Q\tilde{\varepsilon}}{4L} = 3\tilde{\varepsilon}$.

Therefore, in both cases $\mathrm{err}_n(A) \geq 3\tilde{\varepsilon} > \varepsilon$. Repeating the same construction from any other certified algorithm $A'$, we obtain that $\inf_{A'}\left(\mathrm{err}_n(A')\right) > \varepsilon$. Since this has been shown for any $n \leq cS_{\mathrm{C}}(f,\varepsilon)/(1+m_\varepsilon)$, by definition of $\tau$ we can conclude that $\tau > cS_{\mathrm{C}}(f,\varepsilon)/(1+m_\varepsilon)$. $\qquad\square$

Putting together Proposition 1 and Theorem 2 shows that the sample complexity of c.DOO applied to any Lipschitz function $f \in \mathcal{F}_L$ is of order $S_{\mathrm{C}}(f,\varepsilon)$ and no certified algorithm $A$ can do better, *not even if $A$ knows $f$ exactly*. This is a striking difference with the classical non-certified setting in which the best algorithm for each fixed function $f$ has trivially sample complexity 1. In particular, for non-pathological sets $\mathcal{X}$, combining Proposition 1 and Theorem 2 with Theorem 1 gives the following near-tight characterization of the optimal sample complexity of certified algorithms.

**Theorem 3.** *Let $L > 0$ and suppose that Assumption 4 holds with $r_0 > \varepsilon_0/2L$. Then, there exist two constants $c_d, C_d > 0$ such that, for all Lipschitz functions $f \in \mathcal{F}_L$ and any accuracy $\varepsilon \in (0, \varepsilon_0]$, the optimal sample complexity of any certified algorithm $A$ satisfies*

$$\frac{c_d\left(1 - \frac{\mathrm{Lip}(f)}{L}\right)^d}{1 + \lceil\log_2(\varepsilon_0/\varepsilon)\rceil}\int_{\mathcal{X}}\frac{\mathrm{d}\boldsymbol{x}}{\left(f(\boldsymbol{x}^\star) - f(\boldsymbol{x}) + \varepsilon\right)^d} \leq \inf_A \sigma(A, f, \varepsilon)$$

$$\leq \sigma(\mathrm{c.DOO}, f, \varepsilon) \leq C_d \int_{\mathcal{X}}\frac{\mathrm{d}\boldsymbol{x}}{\left(f(\boldsymbol{x}^\star) - f(\boldsymbol{x}) + \varepsilon\right)^d}\ .$$

*For example, one can take $c_d = \frac{L^d}{4\cdot 2^7 d v_1}$ and $C_d = \left(2 + K\left(\mathbf{1}_{\nu/R \geq 1} + \mathbf{1}_{\nu/R < 1}(4R/\nu)^d\right)\right)\frac{(128L)^d}{v_1\gamma}$.*

**The boundary case $L = \mathrm{Lip}(f)$.** The previous results show that, for any function $f$ whose best Lipschitz constant $\mathrm{Lip}(f)$ is bounded away from $L$, the quantity $S_{\mathrm{C}}(f,\varepsilon)$ (or the equivalent integral bound) characterizes the optimal sample complexity up to log terms and dimension-dependent constants. For the sake of completeness, we now discuss the boundary case in which $L = \mathrm{Lip}(f)$, i.e., when the best Lipschitz constant of the target function is known exactly by the algorithm. As we mentioned earlier, this case is not of great relevance in practice, as one can hardly think of a scenario in which $f$ is unknown but the learner has somehow perfect knowledge of its smallest Lipschitz constant $\mathrm{Lip}(f)$. It could, however, be of some theoretical interest.

Next we show an interesting difference between dimensions $d = 1$ and $d \geq 2$. In dimension one, $S_{\mathrm{C}}(f,\varepsilon)$ nearly characterizes the optimal sample complexity even when $L = \mathrm{Lip}(f)$, as shown below. The proof is deferred to Section D.2 of the Supplementary Material.

**Proposition 2.** *If $d = 1$, let $c = 2^{-8}/3$. Then, the sample complexity of any certified algorithm $A$ satisfies, for any $L$-Lipschitz function $f$ and all $\varepsilon \in (0, \varepsilon_0]$, $\sigma(A, f, \varepsilon) > cS_{\mathrm{C}}(f, \varepsilon)/(1 + m_\varepsilon)$.*

In contrast, in dimension $d \geq 2$, there are examples of functions $f$ with $\mathrm{Lip}(f) = L$ for which the optimal sample complexity is much smaller than $S_{\mathrm{C}}(f, \varepsilon)$. In the proposition below, the fact that $\mathrm{Lip}(f) = L$, the specific shape of $f$, and the specific initialization point enable the learner to find *and certify* a maximizer of $f$ in only two rounds, while $S_{\mathrm{C}}(f, \varepsilon)$ grows polynomially in $1/\varepsilon$. The proof is deferred to Section D.3 of the Supplementary Material, together with a refresher on PS.

**Proposition 3.** *Let $d \geq 2$, $\mathcal{X} := B_1$, and $\|\cdot\|$ be a norm. The certified Piyavskii-Shubert algorithm PS with initial guess $\boldsymbol{x}_1 := \boldsymbol{0}$ is a certified algorithm satisfying, for the $L$-Lipschitz function $f := L\|\cdot\|$ and any $\varepsilon \in (0, \varepsilon_0)$, $\sigma(\mathrm{PS}, f, \varepsilon) = 2 \ll c_d/\varepsilon^{d-1} \leq S_{\mathrm{C}}(f, \varepsilon)$, for some constant $c_d > 0$.*

Note that the same upper bound of 2 could be proved in dimension $d = 1$ for $f(x) = L|x|$, but in this case, $S_{\mathrm{C}}(f, \varepsilon)/(1 + m_\varepsilon)$ is of constant order. There is therefore no contradiction with Proposition 2.

Also, remark that Proposition 3 does not solve the question of characterizing the optimal sample complexity for any $f$ such that $\mathrm{Lip}(f) = L$. We conjecture that the drastic improvement in the sample complexity shown in the specific example above is not possible for all other functions $f$ with $\mathrm{Lip}(f) = L$. For instance, we believe that $S_{\mathrm{C}}(f, \varepsilon)$ remains the right quantity for functions for which $|f(x) - f(y)|/\|x - y\|$ is close to $\mathrm{Lip}(f)$ only far away from the set of maximizers.

## 5   Conclusions and Open Problems

**Contributions.** We studied the sample complexity of certified zeroth-order Lipschitz optimization. We first showed that the sample complexity of the computationally tractable c.DOO algorithm scales with the quantity $S_{\mathrm{C}}(f, \varepsilon)$ introduced in Eq. (5) (Proposition 1). We then characterized this quantity in terms of the integral expression $\int_{\mathcal{X}} \mathrm{d}\boldsymbol{x}/(\max(f) - f(\boldsymbol{x}) + \varepsilon)^d$ (Theorem 1), solving as a corollary a long-standing open problem in Hansen et al. (1991). Finally we proved an instance-dependent lower bound (Theorem 2) showing that this integral characterizes (up to $\log$ factors) the optimal sample complexity of certified zeroth-order Lipschitz optimization in any dimensions $d \geq 1$ whenever the smallest Lipschitz constant of $f$ is not known exactly by the algorithm (Theorem 3).

**Limitations and open problems.** There are some interesting directions that would be worth investigating in the future but we did not cover in this paper. First, even if the results of Section 2 could be easily extended to pseudo-metric spaces as in Munos (2014) and related works, our other results are finite-dimensional and exploit the normed space structure.

Second, our lower and upper bounds involve constants (with respect to $f$ and $\varepsilon$) that depend exponentially on $d$. Although the dependency in $d$ was intentionally not optimized here, it would be beneficial to obtain tighter constants. However, we believe that removing the exponential dependency in $d$ altogether, while keeping the same level of generality as this paper, would be very challenging, if at all possible. There is also room for improvement in the specific (and unrealistic) case $L = \mathrm{Lip}(f)$, for which characterizing the optimal sample complexity is still open in dimensions $d \geq 2$.

A third question is related to the general notion of adaptivity to smoothness (e.g., Munos 2011; Bartlett et al. 2019). Note that when no upper bound $L$ on $\mathrm{Lip}(f)$ is available, it is in general not possible to issue a finite certificate $\xi_n$ satisfying $\max(f) - f(\boldsymbol{x}_n^\star) \leq \xi_n$ for any $f$ (as arbitrarily steep bumps can be added to $f$). Hence, while having some upper bound $L$ is necessary for certified optimization, a natural question is whether $L$ could be much larger than $\mathrm{Lip}(f)$ without penalizing significantly the sample complexity. Theorem 2 provides a negative answer since, for a fixed $f$ for which $\boldsymbol{x}^\star$ is in the interior of $\mathcal{X}$, when $L \to \infty$, the lower bound on $\sigma(A, f, \varepsilon)$ is of order $S_C(f, \varepsilon)/\log(1/\varepsilon)$ and $S_C(f, \varepsilon)$ grows like $L^d$. It would thus be interesting to see whether a relaxed version of certification would make sense for cases when only a very coarse upper bound $L$ on $\mathrm{Lip}(f)$ is known.

Finally, it would also be interesting to see if, using randomized algorithms (still with exact observations of $f$) and weakening the notion of certification (requiring it to hold only with some prescribed probability), smaller upper bounds could be obtained. Also, an important question is to quantify the sample complexity increase yielded by having only noisy observations of $f$ (Bubeck et al., 2011; Kleinberg et al., 2019). We expect the modified integral $\int_{\mathcal{X}} \mathrm{d}\boldsymbol{x}/(\max(f) - f(\boldsymbol{x}) + \varepsilon)^{d+2}$ to play a role for this, where our intuition for the "+2" is that to see an $\varepsilon$-big gap through the variance, the learner has to draw roughly $\varepsilon^{-2}$ samples.

## Acknowledgments and Disclosure of Funding

The work of Tommaso Cesari and Sébastien Gerchinovitz has benefitted from the AI Interdisciplinary Institute ANITI, which is funded by the French "Investing for the Future – PIA3" program under the Grant agreement ANR-19-P3IA-0004. Sébastien Gerchinovitz gratefully acknowledges the support of the DEEL project[4]. This work benefited from the support of the project BOLD from the French national research agency (ANR).

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
