# Instance-Dependent Bounds for Zeroth-order Lipschitz Optimization with Error Certificates
## Supplementary Material

**François Bachoc**
Institut de Mathématiques de Toulouse & University Paul Sabatier
`francois.bachoc@math.univ-toulouse.fr`

**Tommaso Cesari**
Toulouse School of Economics
`tommaso-renato.cesari@univ-toulouse.fr`

**Sébastien Gerchinovitz**
IRT Saint Exupéry & Institut de Mathématiques de Toulouse
`sebastien.gerchinovitz@irt-saintexupery.com`

## A  Useful Results on Packing and Covering

For the sake of completeness, we recall the definitions of packing and covering numbers, as well as some known useful results.

**Definition 1.** *Fix any norm $\|\cdot\|$. For any non-empty and bounded subset $E$ of $\mathbb{R}^d$ and all $r > 0$,*

- *the $r$-packing number of $E$ is the largest cardinality of an $r$-packing of $E$, i.e.,*

$$\mathcal{N}(E, r) := \sup\left\{ k \in \mathbb{N}^* : \exists \boldsymbol{x}_1, \ldots, \boldsymbol{x}_k \in E, \min_{i \neq j} \|\boldsymbol{x}_i - \boldsymbol{x}_j\| > r \right\} ;$$

- *the $r$-covering number of $E$ is the smallest cardinality of an $r$-covering of $E$, i.e.,*

$$\mathcal{M}(E, r) := \min\left\{ k \in \mathbb{N}^* : \exists \boldsymbol{x}_1, \ldots, \boldsymbol{x}_k \in \mathbb{R}^d, \forall \boldsymbol{x} \in E, \exists i \in [k], \|\boldsymbol{x} - \boldsymbol{x}_i\| \leq r \right\} .$$

*We also define $\mathcal{N}(\varnothing, r) = \mathcal{M}(\varnothing, r) = 0$ for all $r > 0$.*

Covering numbers and packing numbers are closely related. In particular, the following well-known inequalities hold—see, e.g., (Wainwright, 2019, Lemmas 5.5 and 5.7, with permuted notation of $\mathcal{M}$ and $\mathcal{N}$).[5]

**Lemma 1.** *Fix any norm $\|\cdot\|$. For any bounded set $E \subset \mathbb{R}^d$ and all $r > 0$,*

$$\mathcal{N}(E, 2r) \leq \mathcal{M}(E, r) \leq \mathcal{N}(E, r) . \tag{8}$$

*Furthermore, for any $\delta > 0$ and all $r > 0$,*

$$\mathcal{M}\left(B_\delta, r\right) \leq \left(1 + 2\frac{\delta}{r}\mathbb{I}_{r<\delta}\right)^d . \tag{9}$$

We now state a known lemma about packing numbers at different scales. This is the go-to result for rescaling packing numbers.

---

[5]The definition of $r$-covering number of a subset $E$ of $\mathbb{R}^d$ implied by (Wainwright, 2019, Definition 5.1) is slightly stronger than the one used in our paper, because elements $x_1, \ldots, x_N$ of $r$-covers belong to $E$ rather than just $\mathbb{R}^d$. Even if we do not need it for our analysis, Inequality (9) holds also in this stronger sense.

**Lemma 2.** *Fix any norm $\|\cdot\|$. For any bounded $E \subset \mathbb{R}^d$ and all $0 < r_1 < r_2 < \infty$, we have*

$$\mathcal{N}(E, r_1) \leq \left(4\frac{r_2}{r_1}\right)^d \mathcal{N}(E, r_2) \ .$$

*Proof.* Fix any bounded $E \subset \mathbb{R}^d$ and $0 < r_1 < r_2 < \infty$. Consider an $r_1$-packing $F = \{\boldsymbol{x}_1, \ldots, \boldsymbol{x}_{N_1}\}$ of $E$ with cardinality $N_1 = \mathcal{N}(E, r_1)$. Consider then the following iterative procedure. Let $F_0 = F$ and initialize $k = 1$. While $F_{k-1}$ is non-empty, let $\boldsymbol{y}_k$ be any point in $F_{k-1}$, let $F_k$ be obtained from $F_{k-1}$ by removing the points at $\|\cdot\|$-distance less or equal to $r_2$ from $\boldsymbol{y}_k$ (including $\boldsymbol{y}_k$ itself), and increase $k$ by one. Then this procedure yields an $r_2$-packing of $E$ with cardinality equal to the number of steps (the final value $k_{\text{fin}}$ of $k$). At each step $k$, the balls with radius $r_1/2$ centered at points that are removed at this step are included in the ball with radius $2r_2$ centered at $\boldsymbol{y}_k$. By a volume argument, then, the number of removed points at each step is smaller than or equal to $v_{2r_2}/v_{r_1/2} = (4r_2/r_1)^d$. Hence the total number of steps $k_{\text{fin}}$ is greater than or equal to $\mathcal{N}(E, r_1)(r_1/4r_2)^d$. This concludes the proof since $\mathcal{N}(E, r_2)$ is greater than or equal to the total number of steps $k_{\text{fin}}$. $\square$

## B   Missing Proofs of Section 2

In Section 2, we introduced a certified version of the DOO algorithm. To prove Proposition 1, we adapt and slightly improve (see Remark 2) some of the arguments in Munos (2011), showing that the sample complexity of c.DOO is upper bounded (up to constants) by $S_{\text{C}}(f, \varepsilon)$, defined in Eq. (5).

**Proof of Proposition 1.**   Recall that $f$ is an $L$-Lipschitz function with a global maximizer $\boldsymbol{x}^\star$.

Let us first show that Algorithm 1 is indeed a certified algorithm, that is, $f(\boldsymbol{x}^\star) - f(\boldsymbol{x}_n) \leq \xi_n$ for all $n \geq 1$. Note that $f(\boldsymbol{x}^\star) - f(\boldsymbol{x}_1) \leq LR = \xi_1$, since $f$ is $L$-Lipschitz and $R$ bounds the diameter of $X_{0,0} \supset \mathcal{X}$. So, take any $n \geq 2$. Consider the state of the algorithm after exactly $n$ evaluations of $f$. Let $(h^\star, i^\star)$ correspond to the last time that Line 5 was reached and let $m$ be the total number of evaluations of $f$ made up to that time ($m \leq n$). Then the error certificate is $\xi_n = f(\boldsymbol{x}_{h^\star, i^\star}) + LR\delta^{h^\star} - f(\boldsymbol{x}_n^\star)$. By induction, it is straightforward to show that the union of the cells in $\mathcal{L}_j$ contains $\mathcal{X}$ at all steps $j \in \mathbb{N}^\star$. Therefore, the global maximizer $\boldsymbol{x}^\star$ belongs to a cell $X_{\bar{h}, \bar{i}}$ with $(\bar{h}, \bar{i}) \in \mathcal{L}_m$. We have, using first Line 5 and then Assumption 2 and that $f$ is $L$-Lipschitz,

$$\begin{aligned} f(\boldsymbol{x}_{h^\star, i^\star}) + LR\delta^{h^\star} &\geq f(\boldsymbol{x}_{\bar{h}, \bar{i}}) + LR\delta^{\bar{h}} \\ &\geq f(\boldsymbol{x}^\star) - LR\delta^{\bar{h}} + LR\delta^{\bar{h}} \\ &= f(\boldsymbol{x}^\star) \ . \end{aligned} \tag{10}$$

This shows $\xi_n \geq f(\boldsymbol{x}^\star) - f(\boldsymbol{x}_n^\star)$. Hence Algorithm 1 is a certified algorithm.

We now show the upper bound on $\sigma(\text{c.DOO}, f, \varepsilon)$. Consider the infinite sequence $((h_\ell^\star, i_\ell^\star))_{\ell \in \mathbb{N}^*}$ of the leaves that are successively selected at Line 5 of Algorithm 1. For any leaf $(h, i) \in ((h_\ell^\star, i_\ell^\star))_{\ell \in \mathbb{N}^*}$, let $N_{h,i}$ be the number of evaluations of $f$ made by Algorithm 1 until the leaf $(h, i)$ is selected at Line 5. Define then the stopping time

$$I_\varepsilon = \inf\left\{\ell \in \mathbb{N}^*; f(\boldsymbol{x}_{h_\ell^\star, i_\ell^\star}) + LR\delta^{h_\ell^\star} \leq \max_{i \in [N_{h_\ell^\star, i_\ell^\star}]} f(\boldsymbol{x}_i) + \varepsilon\right\}$$

which corresponds to the first iteration when the event

$$f(\boldsymbol{x}_{h^\star, i^\star}) + LR\delta^{h^\star} \leq \max_{i \in [N_{h^\star, i^\star}]} f(\boldsymbol{x}_i) + \varepsilon \tag{11}$$

holds at Line 5. Consider the $N$-th evaluation of $f$ with $N = N_{h_{I_\varepsilon}^\star, i_{I_\varepsilon}^\star} + 1$, that is, the first evaluation of $f$ after the event (11) holds for the first time. Then we have, from (11) with $(h^\star, i^\star) = (h_{I_\varepsilon}^\star, i_{I_\varepsilon}^\star)$,

$$f(\boldsymbol{x}_{h^\star, i^\star}) + LR\delta^{h^\star} \leq \max_{i \in [N_{h^\star, i^\star}]} f(\boldsymbol{x}_i) + \varepsilon \leq \max_{i \in [N]} f(\boldsymbol{x}_i) + \varepsilon,$$

and thus $\xi_N \leq \varepsilon$. Since by definition $\sigma(\text{c.DOO}, f, \varepsilon) = \min\{n \in \mathbb{N}^* : \xi_n \leq \varepsilon\}$, we have

$$\sigma(\text{c.DOO}, f, \varepsilon) \leq N = N_{h_{I_\varepsilon}^\star, i_{I_\varepsilon}^\star} + 1 \leq 2 + K(I_\varepsilon - 1) \ . \tag{12}$$

We now bound $I_\varepsilon - 1$ from above. Assume without loss of generality that $I_\varepsilon - 1 \geq 1$ and consider the sequence $(h_1^\star, i_1^\star), \ldots, (h_{I_\varepsilon-1}^\star, i_{I_\varepsilon-1}^\star)$ corresponding to the first $I_\varepsilon - 1$ times the DOO algorithm went through Line 5. Let $\mathcal{E}_\varepsilon$ be the corresponding finite set $\{\boldsymbol{x}_{h_1^\star, i_1^\star}, \ldots, \boldsymbol{x}_{h_{I_\varepsilon-1}^\star, i_{I_\varepsilon-1}^\star}\}$. Recall that $\varepsilon_{m_\varepsilon} := \varepsilon$ and $\varepsilon_i := \varepsilon_0 2^{-i}$ for $i < m_\varepsilon$, with $\varepsilon_0 := L \max_{\boldsymbol{x}, \boldsymbol{y} \in \mathcal{X}} \|\boldsymbol{x} - \boldsymbol{y}\|$. Recall also that $\mathcal{X}_\varepsilon := \{\boldsymbol{x} \in \mathcal{X} : \max(f) - f(\boldsymbol{x}) \leq \varepsilon\}$ and for all $0 \leq a < b$, $\mathcal{X}_{(a,b]} := \{\boldsymbol{x} \in \mathcal{X} : a < f(\boldsymbol{x}^\star) - f(\boldsymbol{x}) \leq b\}$. Since any $\boldsymbol{x} \in \mathcal{X}$ is $\varepsilon_0$-optimal, then it either belongs to $\mathcal{X}_\varepsilon$ or one of the layers $\mathcal{X}_{(\varepsilon_i, \varepsilon_{i-1}]}$. Thus we have $\mathcal{E}_\varepsilon \subset \mathcal{X}_\varepsilon \bigcup \left( \bigcup_{i=1}^{m_\varepsilon} \mathcal{X}_{(\varepsilon_i, \varepsilon_{i-1}]} \right)$, so that

$$I_\varepsilon - 1 = \operatorname{card}(\mathcal{E}_\varepsilon) \leq \operatorname{card}\left( \mathcal{E}_\varepsilon \cap \mathcal{X}_\varepsilon \right) + \sum_{i=1}^{m_\varepsilon} \operatorname{card}\left( \mathcal{E}_\varepsilon \cap \mathcal{X}_{(\varepsilon_i, \varepsilon_{i-1}]} \right) . \tag{13}$$

Let $N_{\varepsilon, m_\varepsilon+1}$ be the cardinality of $\mathcal{E}_\varepsilon \cap \mathcal{X}_\varepsilon$. For $i = 1, \ldots, m_\varepsilon$, let $N_{\varepsilon,i}$ be the cardinality of $\mathcal{E}_\varepsilon \cap \mathcal{X}_{(\varepsilon_i, \varepsilon_{i-1}]}$.

Note that the arguments leading to (10) imply that for $\boldsymbol{x}_{h,j} \in \mathcal{E}_\varepsilon$,

$$\boldsymbol{x}_{h,j} \in \mathcal{X}_{LR\delta^h} . \tag{14}$$

Consider two distinct $\boldsymbol{x}_{h,j}, \boldsymbol{x}_{h',j'} \in \mathcal{E}_\varepsilon \cap \mathcal{X}_{(\varepsilon_i, \varepsilon_{i-1}]}$. Then, from Assumption 3 and (14), we obtain

$$\|\boldsymbol{x}_{h,j} - \boldsymbol{x}_{h',j'}\| \geq \nu \delta^{\max(h,h')} > \frac{\nu \varepsilon_i}{LR} .$$

Hence, by definition of packing numbers, we have

$$N_{\varepsilon,i} \leq \mathcal{N}\left( \mathcal{X}_{(\varepsilon_i, \varepsilon_{i-1}]}, \frac{\nu \varepsilon_i}{LR} \right) .$$

Using now Lemma 2 (Section A), we obtain

$$N_{\varepsilon,i} \leq \left( \mathbf{1}_{\nu/R \geq 1} + \mathbf{1}_{\nu/R < 1} \left( \frac{4R}{\nu} \right)^d \right) \mathcal{N}\left( \mathcal{X}_{(\varepsilon_i, \varepsilon_{i-1}]}, \frac{\varepsilon_i}{L} \right) . \tag{15}$$

Let now $\boldsymbol{x}_{h_\ell^\star, i_\ell^\star} \in \mathcal{E}_\varepsilon \cap \mathcal{X}_\varepsilon$, with $\ell \in \{1, \ldots, I_\varepsilon - 1\}$. The leaf $(h_\ell^\star, i_\ell^\star)$ was selected when the algorithm went through Line 5 for the $\ell$-th time. By definition of $I_\varepsilon$, the event (11) does not hold when $(h^\star, i^\star) = (h_\ell^\star, i_\ell^\star)$ and thus

$$f(\boldsymbol{x}_{h_\ell^\star, i_\ell^\star}) + LR\delta^{h_\ell^\star} > \max_{i \in [N_{h_\ell^\star, i_\ell^\star}]} f(\boldsymbol{x}_i) + \varepsilon \geq f(\boldsymbol{x}_{h_\ell^\star, i_\ell^\star}) + \varepsilon .$$

This implies that $LR\delta^{h_\ell^\star} > \varepsilon$ and thus

$$\delta^{h_\ell^\star} > \frac{\varepsilon}{LR} .$$

Now consider two distinct $\boldsymbol{x}_{h,j}, \boldsymbol{x}_{h',j'} \in \mathcal{E}_\varepsilon \cap \mathcal{X}_\varepsilon$. Then, from Assumption 3, we obtain

$$\|\boldsymbol{x}_{h,j} - \boldsymbol{x}_{h',j'}\| \geq \nu \delta^{\max(h,h')} > \frac{\nu \varepsilon}{LR} .$$

Hence, we have

$$N_{\varepsilon, m_\varepsilon+1} \leq \mathcal{N}\left( \mathcal{X}_\varepsilon, \frac{\nu \varepsilon}{LR} \right) .$$

Using now Lemma 2 (Section A), we obtain

$$N_{\varepsilon, m_\varepsilon+1} \leq \left( \mathbf{1}_{\nu/R \geq 1} + \mathbf{1}_{\nu/R < 1} \left( \frac{4R}{\nu} \right)^d \right) \mathcal{N}\left( \mathcal{X}_\varepsilon, \frac{\varepsilon}{L} \right) .$$

Combining (12) and (13) with (15) and the last inequality concludes the proof. $\qquad \square$

**Remark 2.** *The analysis of the DOO algorithm in Munos (2011, Theorem 1) does not address the certified setting. The previous proof adapts this analysis to the certified setting and, in passing, slightly improves some of the arguments. Indeed, when counting the cell representatives that are selected, Munos (2011, Theorem 1) partitions the domain $\mathcal{X}$ at any depth $h$ of the tree, yielding bounds involving packing numbers of the form $\mathcal{N}\left( \mathcal{X}_{\varepsilon_{k-1}}, \frac{\varepsilon_k}{L} \right)$, $k = 1, \ldots, m_\varepsilon$. In contrast we partition the values of $f$, yielding bounds involving the smaller packing numbers $\mathcal{N}\left( \mathcal{X}_{(\varepsilon_k, \varepsilon_{k-1}]}, \frac{\varepsilon_k}{L} \right)$, $k = 1, \ldots, m_\varepsilon$ (and $\mathcal{N}\left( \mathcal{X}_\varepsilon, \frac{\varepsilon}{L} \right)$ that is specific to the certified setting). This improvement also enables us to slightly refine the bound of Munos (2011, Theorem 1) in the non-certified setting, see Remark 4 in Section E. We also refer to this remark for more details on the two partitions just discussed.*

**Remark 3.** *The bound of Proposition 1, based on Eq. (5), is built by partitioning $[0, \varepsilon_0]$ into the $m_\varepsilon + 1$ sets $[0, \varepsilon], (\varepsilon, \varepsilon_{m_\varepsilon - 1}], (\varepsilon_{m_\varepsilon - 1}, \varepsilon_{m_\varepsilon - 2}], \ldots, (\varepsilon_1, \varepsilon_0]$ whose lengths are sequentially doubled (except from $[0, \varepsilon]$ to $(\varepsilon, \varepsilon_{m_\varepsilon - 1}]$ and from $(\varepsilon, \varepsilon_{m_\varepsilon - 1}]$ to $(\varepsilon_{m_\varepsilon - 1}, \varepsilon_{m_\varepsilon - 2}])$. As can be seen from the proof of Proposition 1, more general bounds could be obtained, based on more general partitions of $[0, \varepsilon_0]$. The benefits of the present partition are the following. First, except for $[0, \varepsilon]$, it considers sets whose upper values are no more than twice the lower values, which controls the magnitude of their corresponding packing numbers in Eq. (5), at scale the lower values. Second, the number of sets in the partition is logarithmic in $1/\varepsilon$ which controls the sum in Eq. (5). Finally, the upper bound is then tight up to a logarithmic factor for functions $f \in \mathcal{F}_L$, as proved in Section 4. Note also that the same generalization could be applied in the non-certified setting, see Section E.*

## C   Missing Proofs of Section 3

We now prove a result on the sum of volumes of overlapping layers that is used in the proof of Theorem 1.

**Lemma 3.** *If $f$ is L-Lipschitz, fix $\varepsilon \in (0, \varepsilon_0]$ and recall that $m_\varepsilon := \lceil \log_2(\varepsilon_0/\varepsilon) \rceil$, $\varepsilon_{m_\varepsilon} := \varepsilon$, and for all $k \leq m_\varepsilon - 1$, $\varepsilon_k =: \varepsilon_0 2^{-k}$. Then*

$$
\frac{\mathrm{vol}(\mathcal{X}_{2\varepsilon})}{\varepsilon^d} + \sum_{k=1}^{m} \frac{\mathrm{vol}\left(\mathcal{X}_{(\frac{1}{2}\varepsilon_k,\, 2\varepsilon_{k-1}]}\right)}{\varepsilon_{k-1}^d} \leq 8^d \left( \frac{\mathrm{vol}(\mathcal{X}_\varepsilon)}{\varepsilon^d} + \sum_{i=1}^{m} \frac{\mathrm{vol}\left(\mathcal{X}_{(\varepsilon_k,\, \varepsilon_{k-1}]}\right)}{\varepsilon_{k-1}^d} \right).
$$

*Proof.* To avoid clutter, we denote $m_\varepsilon$ simply by $m$. Assume first that $m \geq 3$. Then, the left hand side can be upper bounded by

$$
\frac{\mathrm{vol}(\mathcal{X}_\varepsilon) + \mathrm{vol}(\mathcal{X}_{(\varepsilon_m,\, \varepsilon_{m-1}]}) + \mathrm{vol}(\mathcal{X}_{(\varepsilon_{m-1},\, \varepsilon_{m-2}]})}{\varepsilon^d}
$$

$$
+ \sum_{k=1}^{m-2} \frac{\mathrm{vol}\left(\mathcal{X}_{(\varepsilon_{k+1},\, \varepsilon_k]}\right) + \mathrm{vol}\left(\mathcal{X}_{(\varepsilon_k,\, \varepsilon_{k-1}]}\right) \mathrm{vol}\left(\mathcal{X}_{(\varepsilon_{k-1},\, \varepsilon_{k-2}]}\right)}{\varepsilon_{k-1}^d}
$$

$$
+ \frac{\mathrm{vol}\left(\mathcal{X}_\varepsilon\right) + \mathrm{vol}\left(\mathcal{X}_{(\varepsilon_m,\, \varepsilon_{m-1}]}\right) + \mathrm{vol}\left(\mathcal{X}_{(\varepsilon_{m-1},\, \varepsilon_{m-2}]}\right) + \mathrm{vol}\left(\mathcal{X}_{(\varepsilon_{m-2},\, \varepsilon_{m-3}]}\right)}{\varepsilon_{m-2}^d}
$$

$$
+ \frac{\mathrm{vol}\left(\mathcal{X}_\varepsilon\right) + \mathrm{vol}\left(\mathcal{X}_{(\varepsilon_m,\, \varepsilon_{m-1}]}\right) + \mathrm{vol}\left(\mathcal{X}_{(\varepsilon_{m-1},\, \varepsilon_{m-2}]}\right)}{\varepsilon_{m-1}^d}
$$

$$
\leq 3\frac{\mathrm{vol}(\mathcal{X}_\varepsilon)}{\varepsilon^d} + (2^d + 2)\frac{\mathrm{vol}(\mathcal{X}_{(\varepsilon_m,\, \varepsilon_{m-1}]})}{\varepsilon_{m-1}^d} + (4^d + 2^d + 1)\frac{\mathrm{vol}(\mathcal{X}_{(\varepsilon_{m-1},\, \varepsilon_{m-2}]})}{\varepsilon_{m-2}^d}
$$

$$
+ \frac{1}{2^d} \sum_{k=2}^{m-1} \frac{\mathrm{vol}\left(\mathcal{X}_{(\varepsilon_k,\, \varepsilon_{k-1}]}\right)}{\varepsilon_{k-1}^d} + \sum_{k=1}^{m-2} \frac{\mathrm{vol}\left(\mathcal{X}_{(\varepsilon_k,\, \varepsilon_{k-1}]}\right)}{\varepsilon_{k-1}^d} + 2^d \sum_{k=1}^{m-3} \frac{\mathrm{vol}\left(\mathcal{X}_{(\varepsilon_k,\, \varepsilon_{k-1}]}\right)}{\varepsilon_{k-1}^d}
$$

$$
= 3\frac{\mathrm{vol}(\mathcal{X}_\varepsilon)}{\varepsilon^d} + \frac{\mathrm{vol}(\mathcal{X}_{(\varepsilon_m,\, \varepsilon_{m-1}]})}{\varepsilon_{m-1}^d} + 4^d\frac{\mathrm{vol}(\mathcal{X}_{(\varepsilon_{m-1},\, \varepsilon_{m-2}]})}{\varepsilon_{m-2}^d}
$$

$$
+ \frac{1}{2^d} \sum_{k=2}^{m-1} \frac{\mathrm{vol}\left(\mathcal{X}_{(\varepsilon_k,\, \varepsilon_{k-1}]}\right)}{\varepsilon_{k-1}^d} + \sum_{k=1}^{m} \frac{\mathrm{vol}\left(\mathcal{X}_{(\varepsilon_k,\, \varepsilon_{k-1}]}\right)}{\varepsilon_{k-1}^d} + 2^d \sum_{k=1}^{m} \frac{\mathrm{vol}\left(\mathcal{X}_{(\varepsilon_k,\, \varepsilon_{k-1}]}\right)}{\varepsilon_{k-1}^d}
$$

where we applied several times the definition of the $\varepsilon_k$'s, the inequality follows by $1/\varepsilon^d + 1/\varepsilon_{m-1}^d + 1/\varepsilon_{m-2}^d \leq \min\{3(1/\varepsilon^d),\, (2^d + 2)(1/\varepsilon_{m-1}^d),\, (4^d + 2^d + 1)(1/\varepsilon_{m-2}^d)\}$, and the bound follows after observing that $\max(3, 1, 4^d) = 4^d$ and $4^d + 1/2^d + 1 + 2^d \leq 8^d$. The simple cases $m = 1$ and $m = 2$ can be treated similarly.  □

We denote by $A + B$ the Minkowski sum of two sets $A, B$ and for any set $A$ and all $\lambda \in \mathbb{R}$, we let $\lambda A := \{\lambda \boldsymbol{a} : \boldsymbol{a} \in A\}$.

**Proposition 4.** *If $f$ is $L$-Lipschitz and $\mathcal{X}$ satisfies Assumption 4 with $r_0 > 0, \gamma \in (0, 1]$, then, for all $0 < w < u < 2Lr_0$,*

$$\mathcal{N}\left(\mathcal{X}_u, \frac{u}{L}\right) \leq \frac{1}{\gamma} \frac{\mathrm{vol}\left(\mathcal{X}_{(3/2)u}\right)}{\mathrm{vol}\left(\frac{u}{2L}B_1\right)} \qquad and \qquad \mathcal{N}\left(\mathcal{X}_{(w,u]}, \frac{w}{L}\right) \leq \frac{1}{\gamma} \frac{\mathrm{vol}\left(\mathcal{X}_{(w/2, 3u/2]}\right)}{\mathrm{vol}\left(\frac{w}{2L}B_1\right)} .$$

*Proof.* Fix any $u > w > 0$. Let $\eta_1 := \frac{u}{L}$, $\eta_2 := \frac{w}{L}$, $E_1 := \mathcal{X}$, $E_2 := \mathcal{X}_w^c$, and $i \in [2]$. Note that for any $\eta > 0$ and $A \subset \mathcal{X}$, the balls of radius $\eta/2$ centered at the elements of an $\eta$-packing of $A$ intersected with $\mathcal{X}$ are all disjoint and included in $\left(A + B_{\eta/2}(\mathbf{0})\right) \cap \mathcal{X}$. Thus, letting $P_i$ be a set of $\eta_i$-separated points included in $A_i := \mathcal{X}_u \cap E_i$ with cardinality $|P_i| = \mathcal{N}(A_i, \eta_i)$, we have

$$\mathrm{vol}\left(\left(A_i + B_{\eta_i/2}(\mathbf{0})\right) \cap \mathcal{X}\right) \geq \sum_{\boldsymbol{x} \in P_i} \mathrm{vol}\left(B_{\eta_i/2}(\boldsymbol{x}) \cap \mathcal{X}\right) \geq \gamma \mathrm{vol}\left(B_{\eta_i/2}(\mathbf{0})\right) \mathcal{N}(A_i, \eta_i) ,$$

where the second inequality follows by Assumption 4. We now further upper bound the left-hand side. Take an arbitrary point $\boldsymbol{x}_i \in (\mathcal{X}_u \cap E_i + B_{\eta_i/2}) \cap \mathcal{X}$. By definition of Minkowski sum, there exists $\boldsymbol{x}_i' \in \mathcal{X}_u \cap E_i$ such that $\|\boldsymbol{x}_i - \boldsymbol{x}_i'\| \leq \eta_i/2$. Hence $f(\boldsymbol{x}^\star) - f(\boldsymbol{x}_i) \leq f(\boldsymbol{x}^\star) - f(\boldsymbol{x}_i') + |f(\boldsymbol{x}_i') - f(\boldsymbol{x}_i)| \leq u + L(\eta_i/2) \leq (3/2)u$. This implies that $\boldsymbol{x}_i \in \mathcal{X}_{(3/2)u}$, which proves the first inequality. For the second one, note that $\boldsymbol{x}_2$ satisfies $f(\boldsymbol{x}^\star) - f(\boldsymbol{x}_2) \geq f(\boldsymbol{x}^\star) - f(\boldsymbol{x}_2') - |f(\boldsymbol{x}_2') - f(\boldsymbol{x}_2)| \geq w - L(\eta_2/2) = (1/2)w$. $\qquad\square$

# D   Missing Proofs of Section 4

In this section we provide all missing details and proofs from Section 4.

## D.1   Missing details in the Proof of Theorem 2

We claimed that the quantity $\tau$ introduced in Eq. (7) lower bounds the sample complexity $\sigma(A, f, \varepsilon)$ of any certified algorithm $A$. To prove this formally, fix an arbitrary certified algorithm $A$, let $N = \sigma(A, f, \varepsilon)$, and assume by contradiction that $N < \tau$. Then we have $\mathrm{err}_N(A) \geq \inf_{A'}\left(\mathrm{err}_N(A')\right) > \varepsilon$ by definition of $\tau$. This means that there exists an $L$-Lipschitz function $g$, coinciding with $f$ on $\boldsymbol{x}_1, \ldots, \boldsymbol{x}_N$ and such that $\max(g) - \boldsymbol{x}_N^\star > \varepsilon$. Now, since $\boldsymbol{x}_i, \boldsymbol{x}_i^\star$, and $\xi_i$ are deterministic functions of the previous observations $f(\boldsymbol{x}_1) = g(\boldsymbol{x}_1), \ldots, f(\boldsymbol{x}_{i-1}) = g(\boldsymbol{x}_{i-1})$ (for all $i = 1, \ldots, N$), running $A$ on either $f$ or $g$ returns the same $\boldsymbol{x}_i, \boldsymbol{x}_i^\star$, and $\xi_i$ (for all $i = 1, \ldots, N$). Thus we have that $\sigma(A, g, \varepsilon) = \sigma(A, f, \varepsilon) = N$. This, together with the fact that $A$ is a certified algorithm, implies that $\varepsilon < \max(g) - \boldsymbol{x}_N^\star \leq \xi_N \leq \varepsilon$, which yields a contradiction. $\qquad\square$

## D.2   Proof of Proposition 2

Let $f$ be an arbitrary $L$-Lipschitz function. Let $Q = 8$. As for the proof of Theorem 2, it is sufficient to show that $\tau > cS_{\mathrm{C}}(f, \varepsilon)/(1 + m_\varepsilon)$, with $\tau$ defined in (7). If $cS_{\mathrm{C}}(f, \varepsilon)/(1 + m_\varepsilon) < 1$, then the result follows by $\tau \geq 1$. Consider then from now on that $cS_{\mathrm{C}}(f, \varepsilon)/(1 + m_\varepsilon) \geq 1$.

Defining $\tilde{\varepsilon}$ as in the proof of Theorem 2, one can prove similarly that $cS_{\mathrm{C}}(f, \varepsilon)/(1 + m_\varepsilon) \leq c\mathcal{N}(\mathcal{X}_{\tilde{\varepsilon}}, \tilde{\varepsilon}/2L)$. From Lemma 2,

$$\mathcal{N}\left(\mathcal{X}_{\tilde{\varepsilon}}, \frac{Q\tilde{\varepsilon}}{L}\right) \geq \frac{1}{8Q}\mathcal{N}\left(\mathcal{X}_{\tilde{\varepsilon}}, \frac{\tilde{\varepsilon}}{2L}\right) \geq \frac{1}{8Q}\frac{S_{\mathrm{C}}(f, \varepsilon)}{m_\varepsilon + 1} \geq 12 ,$$

because $c = 1/96Q$ and $cS_{\mathrm{C}}(f, \varepsilon)/(1 + m_\varepsilon) \geq 1$. Let now $n \leq cS_{\mathrm{C}}(f, \varepsilon)/(1 + m_\varepsilon)$. Then we have $n \leq c(8Q)\mathcal{N}(\mathcal{X}_{\tilde{\varepsilon}}, Q\tilde{\varepsilon}/L)$. Thus, by $c(8Q) = 1/12$, $n \leq \mathcal{N}(\mathcal{X}_{\tilde{\varepsilon}}, Q\tilde{\varepsilon}/L)/12$, and $\mathcal{N}(\mathcal{X}_{\tilde{\varepsilon}}, Q\tilde{\varepsilon}/L) \geq 12$, we have

$$n \leq \left\lceil \frac{\mathcal{N}\left(\mathcal{X}_{\tilde{\varepsilon}}, \frac{Q\tilde{\varepsilon}}{L}\right)}{2} \right\rceil - 4 . \tag{16}$$

Consider a certified algorithm $A$ for $L$-Lipschitz functions. Let us consider a $Q\tilde{\varepsilon}/L$ packing $\tilde{x}_1 < \tilde{x}_2 < \cdots < \tilde{x}_N$ of $\mathcal{X}_{\tilde{\varepsilon}}$ with $N = \mathcal{N}(\mathcal{X}_{\tilde{\varepsilon}}, Q\tilde{\varepsilon}/L)$. Consider the $\lfloor N/2 \rfloor - 1$ disjoint open segments $(\tilde{x}_1, \tilde{x}_3)$, $(\tilde{x}_3, \tilde{x}_5)$, ..., $(\tilde{x}_{2\lfloor N/2 \rfloor - 3}, \tilde{x}_{2\lfloor N/2 \rfloor - 1})$. Then from (16) there exists $i \in \{1, 3, \ldots, 2\lfloor N/2 \rfloor - 3\}$ such that the segment $(\tilde{x}_i, \tilde{x}_{i+2})$ does not contain any of

the points $x_1 = x_1(A, f), \ldots, x_n = x_n(A, f)$ that $A$ queries when run on $f$. Assume that $\tilde{x}_{i+1} - \tilde{x}_i \leq \tilde{x}_{i+2} - \tilde{x}_{i+1}$ (the case $\tilde{x}_{i+1} - \tilde{x}_i > \tilde{x}_{i+2} - \tilde{x}_{i+1}$ can be treated analogously; we omit these straightforward details for the sake of conciseness). Consider the function $h_{+,\tilde{\varepsilon}} \colon \mathcal{X} \to \mathbb{R}$ defined by

$$h_{+,\tilde{\varepsilon}}(x) = \begin{cases} f(x) & \text{if } x \in \mathcal{X} \backslash [\tilde{x}_i, \tilde{x}_{i+2}] \\ f(\tilde{x}_i) + L(x - \tilde{x}_i) & \text{if } x \in \mathcal{X} \cap [\tilde{x}_i, \tilde{x}_{i+1}] \\ f(\tilde{x}_i) + L(\tilde{x}_{i+1} - \tilde{x}_i) + (x - \tilde{x}_{i+1})\frac{f(\tilde{x}_{i+2}) - f(\tilde{x}_i) - L(\tilde{x}_{i+1} - \tilde{x}_i)}{\tilde{x}_{i+2} - \tilde{x}_{i+1}} & \text{if } x \in \mathcal{X} \cap (\tilde{x}_{i+1}, \tilde{x}_{i+2}] \, . \end{cases}$$

We see that $h_{+,\tilde{\varepsilon}}$ is $L$-Lipschitz (since $\tilde{x}_{i+1} - \tilde{x}_i \leq \tilde{x}_{i+2} - \tilde{x}_{i+1}$). Furthermore, $h_{+,\tilde{\varepsilon}}$ coincides with $f$ at all query points $x_1, \ldots, x_n$. Similarly, consider the function $h_{-,\tilde{\varepsilon}} \colon \mathcal{X} \to \mathbb{R}$ defined by

$$h_{-,\tilde{\varepsilon}}(x) = \begin{cases} f(x) & \text{if } x \in \mathcal{X} \backslash [\tilde{x}_i, \tilde{x}_{i+2}] \\ f(\tilde{x}_i) - L(x - \tilde{x}_i) & \text{if } x \in \mathcal{X} \cap [\tilde{x}_i, \tilde{x}_{i+1}] \\ f(\tilde{x}_i) - L(\tilde{x}_{i+1} - \tilde{x}_i) + (x - \tilde{x}_{i+1})\frac{f(\tilde{x}_{i+2}) - f(\tilde{x}_i) + L(\tilde{x}_{i+1} - \tilde{x}_i)}{\tilde{x}_{i+2} - \tilde{x}_{i+1}} & \text{if } x \in \mathcal{X} \cap (\tilde{x}_{i+1}, \tilde{x}_{i+2}] \, . \end{cases}$$

As before, $h_{-,\tilde{\varepsilon}}$ is $L$-Lipschitz and coincides with $f$ on $x_1, \ldots, x_n$.

Let $x_n^\star = x_n^\star(A, f)$ be the recommendation of $A$ at round $n$ when run on $f$.

Case 1: $x_n^\star \in \mathcal{X} \backslash [\tilde{x}_i, \tilde{x}_{i+2}]$. Then, since $\tilde{x}_i \in \mathcal{X}_{\tilde{\varepsilon}}$ and $\tilde{x}_{i+1} - \tilde{x}_i \geq Q\tilde{\varepsilon}/L$, we have

$$h_{+,\tilde{\varepsilon}}(\tilde{x}_{i+1}) - h_{+,\tilde{\varepsilon}}(x_n^\star) = f(\tilde{x}_i) + L(\tilde{x}_{i+1} - \tilde{x}_i) - f(x_n^\star) \geq -\tilde{\varepsilon} + L\frac{Q\tilde{\varepsilon}}{L} = 7\tilde{\varepsilon} \, .$$

Case 2: $x_n^\star \in \mathcal{X} \cap \left[\tilde{x}_i, (\tilde{x}_i + \tilde{x}_{i+1})/2\right]$. Then, since $\tilde{x}_{i+1} - \tilde{x}_i \geq Q\tilde{\varepsilon}/L$, we have

$$h_{+,\tilde{\varepsilon}}(\tilde{x}_{i+1}) - h_{+,\tilde{\varepsilon}}(x_n^\star) = f(\tilde{x}_i) + L(\tilde{x}_{i+1} - \tilde{x}_i) - f(\tilde{x}_i) - L(x_n^\star - \tilde{x}_i) \geq L\frac{\tilde{x}_{i+1} - \tilde{x}_i}{2} \geq 4\tilde{\varepsilon} \, .$$

Case 3: $x_n^\star \in \mathcal{X} \cap \left[(\tilde{x}_i + \tilde{x}_{i+1})/2, \tilde{x}_{i+1}\right]$. Then, since $\tilde{x}_{i+1} - \tilde{x}_i \geq Q\tilde{\varepsilon}/L$, we have

$$h_{-,\tilde{\varepsilon}}(\tilde{x}_i) - h_{-,\tilde{\varepsilon}}(x_n^\star) = f(\tilde{x}_i) - f(\tilde{x}_i) + L(x_n^\star - \tilde{x}_i) \geq L\frac{\tilde{x}_{i+1} - \tilde{x}_i}{2} \geq 4\tilde{\varepsilon} \, .$$

Case 4: $x_n^\star \in \mathcal{X} \cap \left[\tilde{x}_{i+1}, (\tilde{x}_{i+1} + \tilde{x}_{i+2})/2\right]$. Then, since $\tilde{x}_{i+1} - \tilde{x}_i \geq Q\tilde{\varepsilon}/L$, since $\tilde{x}_i, \tilde{x}_{i+2} \in \mathcal{X}_{\tilde{\varepsilon}}$, and since $h_{-,\tilde{\varepsilon}}$ is linear increasing on $[\tilde{x}_{i+1}, \tilde{x}_{i+2}]$ with left value $f(\tilde{x}_i) - L(\tilde{x}_{i+1} - \tilde{x}_i)$ and right value $f(\tilde{x}_{i+2})$, we have

$$h_{-,\tilde{\varepsilon}}(\tilde{x}_i) - h_{-,\tilde{\varepsilon}}(x_n^\star) \geq f(\tilde{x}_i) - \frac{f(\tilde{x}_i) - L(\tilde{x}_{i+1} - \tilde{x}_i) + f(\tilde{x}_{i+2})}{2}$$
$$= \frac{f(\tilde{x}_i) - f(\tilde{x}_{i+2})}{2} + L\frac{\tilde{x}_{i+1} - \tilde{x}_i}{2} \geq -\frac{\tilde{\varepsilon}}{2} + \frac{Q}{2}\tilde{\varepsilon} \geq 3\tilde{\varepsilon} \, .$$

Case 5: $x_n^\star \in \mathcal{X} \cap \left[(\tilde{x}_{i+1} + \tilde{x}_{i+2})/2, \tilde{x}_{i+2}\right]$. Then, since $\tilde{x}_{i+1} - \tilde{x}_i \geq Q\tilde{\varepsilon}/L$, since $\tilde{x}_i, \tilde{x}_{i+2} \in \mathcal{X}_{\tilde{\varepsilon}}$ and since $h_{+,\tilde{\varepsilon}}$ is linear decreasing on $[\tilde{x}_{i+1}, \tilde{x}_{i+2}]$ with left value $f(\tilde{x}_i) + L(\tilde{x}_{i+1} - \tilde{x}_i)$ and right value $f(\tilde{x}_{i+2})$, we have

$$h_{+,\tilde{\varepsilon}}(\tilde{x}_{i+1}) - h_{+,\tilde{\varepsilon}}(x_n^\star) \geq f(\tilde{x}_i) + L(\tilde{x}_{i+1} - \tilde{x}_i) - \frac{f(\tilde{x}_i) + L(\tilde{x}_{i+1} - \tilde{x}_i) + f(\tilde{x}_{i+2})}{2}$$
$$= \frac{f(\tilde{x}_i) - f(\tilde{x}_{i+2})}{2} + L\frac{\tilde{x}_{i+1} - \tilde{x}_i}{2} \geq -\frac{\tilde{\varepsilon}}{2} + \frac{Q}{2}\tilde{\varepsilon} \geq 3\tilde{\varepsilon} \, .$$

Putting all cases together and recalling the definition of $\mathrm{err}_n(A)$ in the proof of Theorem 2, we then obtain $\mathrm{err}_n(A) \geq 3\tilde{\varepsilon} > \varepsilon$. Being $A$ arbitrary, this implies $\inf_{A'} \mathrm{err}_n(A') > \varepsilon$. Since this has been shown for any $n \leq cS_{\mathrm{C}}(f, \varepsilon)/(1 + m_\varepsilon)$ we thus have $\tau > cS_{\mathrm{C}}(f, \varepsilon)/(1 + m_\varepsilon)$. $\qquad \square$

## D.3 The Piyavskii-Shubert Algorithm and Proof of Proposition 3

**The Piyavskii-Shubert Algorithm.** In this section, we recall the definition of the certified Piyavskii-Shubert algorithm (Algorithm 2, Piyavskii 1972; Shubert 1972) and we show that if $\mathrm{Lip}(f) = L$ (i.e., if the best Lipschitz constant of $f$ is known exactly by the algorithm) the sample complexity can be constant in dimension $d \geq 2$ (Proposition 3).

---
**Algorithm 2:** Certified Piyavskii-Shubert algorithm (PS)
---
**input:** Lipschitz constant $L > 0$, norm $\|\cdot\|$, initial guess $\boldsymbol{x}_1 \in \mathcal{X}$

**for** $i = 1, 2, \ldots$ **do**

    pick the next query point $\boldsymbol{x}_i$

    observe the value $f(\boldsymbol{x}_i)$

    output the recommendation $\boldsymbol{x}_i^\star \leftarrow \operatorname{argmax}_{\boldsymbol{x} \in \{\boldsymbol{x}_1, \ldots, \boldsymbol{x}_i\}} f(\boldsymbol{x})$

    output the error certificate $\xi_i = \widehat{f}_i^\star - f_i^\star$, where $\widehat{f}_i(\cdot) \leftarrow \min_{j \in [i]} \{f(\boldsymbol{x}_j) + L \|\boldsymbol{x}_j - (\cdot)\|\}$,

    $\widehat{f}_i^\star \leftarrow \max_{\boldsymbol{x} \in \mathcal{X}} \widehat{f}_i(\boldsymbol{x})$, $f_i^\star \leftarrow \max_{j \in [i]} f(\boldsymbol{x}_j)$, and let $\boldsymbol{x}_{i+1} \in \operatorname{argmax}_{\boldsymbol{x} \in \mathcal{X}} \widehat{f}_i(\boldsymbol{x})$
---

**Proof of Proposition 3.** Fix any $\varepsilon \in (0, \varepsilon_0)$ and any $L$-Lipschitz function $f$. Since $f$ is $L$-Lipschitz, then $\max_{\boldsymbol{x} \in \mathcal{X}} \widehat{f}_i(\boldsymbol{x}) \geq \max_{\boldsymbol{x} \in \mathcal{X}} f(\boldsymbol{x})$ for all $i \in \mathbb{N}^*$. Hence $\max_{\boldsymbol{x} \in \mathcal{X}} f(x) - f(\boldsymbol{x}_i^\star) \leq \max_{\boldsymbol{x} \in \mathcal{X}} \widehat{f}_i(\boldsymbol{x}) - f_i^\star = \xi_i$. This shows that the certified Piyavskii-Shubert algorithm is indeed a certified algorithm. Then, if $f := L \|\cdot\|$ and $\boldsymbol{x}_1 := \boldsymbol{0}$, we have that $\widehat{f}_1 = f$, $\xi_1 = L$, and $\boldsymbol{x}_2$ belongs to the the unit sphere, i.e., $\boldsymbol{x}_2$ is a maximizer of $f$. Since $\widehat{f}_2 = f$, we have that $\xi_2 = 0$, hence $\sigma(\mathrm{PS}, f, \varepsilon) = 2$. Finally, by definition (5), we have $S_{\mathrm{C}}(f, \varepsilon) \geq \mathcal{N}(\operatorname{argmax}_{\mathcal{X}} f, \varepsilon/L)$. Since $\operatorname{argmax}_{\mathcal{X}} f$ is the unit sphere, there exists a constant $c_d$, only depending on $d$, $\|\cdot\|$ and $L$, such that $S_{\mathrm{C}}(f, \varepsilon) \geq c_d / \varepsilon^{d-1}$. $\qquad\square$

We give some intuition on Proposition 3. Consider a function $f$ that has Lipschitz constant exactly $L$, and a pair of points in $\mathcal{X}$ whose respective values of $f$ are maximally distant, that is the difference of values of $f$ is exactly $L$ times the norm of the input difference. This configuration provides strong information on the value of the global maximum of $f$, as is illustrated in the proof of Proposition 3. Another interpretation is that when $f$ has Lipschitz constant exactly $L$, there is less flexibility for the $L$-Lipschitz function $g$ that yields the maximal optimization error in $\mathrm{err}_n(A)$ (introduced in the proof of Theorem 2).

## E    Comparison with the classical non-certified setting

For the interested reader who is not familiar with DOO, in this section, we recall and analyze the classical non-certified version of this algorithm. As mentioned in Remark 2, our analysis is slightly tighter than that of Munos (2011), and serves as a better comparison for highlighting the differences between the certified and the non-certified settings (see Remark 5 below).

The difference between our certified version c.DOO and the classical non-certified DOO algorithm (denoted by nc.DOO below) is that the latter does not output any certificates $\xi_1, \xi_2, \ldots$. In other words, nc.DOO coincides with Algorithm 1 except for Lines 3 and 13. In particular, it outputs the same query points $\boldsymbol{x}_1, \boldsymbol{x}_2, \ldots$ and recommendations $\boldsymbol{x}_1^\star, \boldsymbol{x}_2^\star, \ldots$ as c.DOO. The performance of this non-certified algorithm is classically measured by the non-certified sample complexity (6), i.e., the smallest number of queries needed before outputting an $\varepsilon$-optimal recommendation.

**Proposition 5.** *If Assumptions 2 and 3 hold, the non-certified sample complexity of the non-certified DOO algorithm* nc.DOO *satisfies, for all Lipschitz functions* $f \in \mathcal{F}_L$[6] *and any accuracy* $\varepsilon \in (0, \varepsilon_0]$,

$$\zeta(\mathrm{nc.DOO}, f, \varepsilon) \leq 1 + C_d \sum_{k=1}^{m_\varepsilon} \mathcal{N}\left(\mathcal{X}_{(\varepsilon_k, \varepsilon_{k-1})}, \frac{\varepsilon_k}{L}\right) \, ,$$

*where* $C_d = K\left(\mathbf{1}_{\nu/R \geq 1} + \mathbf{1}_{\nu/R < 1}(4R/\nu)^d\right)$.

*Proof.* The proof of Proposition 1 (Section B), from the beginning to (10), implies that, for any $(h^\star, i^\star)$ in Line 5 of Algorithm 1,

$$f(\boldsymbol{x}_{h^\star, i^\star}) \in \mathcal{X}_{LR\delta^{h^\star}}. \tag{17}$$

The guarantee (17) is classical (e.g., Munos 2011).

We now proceed in a direction that is slightly different from the proof of Munos (2011, Theorem 1). Consider the first time at which the DOO algorithm reaches Line 5 with $f(\boldsymbol{x}_{h^\star, i^\star}) \geq f(\boldsymbol{x}^\star) - \varepsilon$.

---

[6]Our proof can be easily adapted to the weaker assumption that $f$ is only $L$-Lipschitz around a maximizer.

Then let $I_\varepsilon$ be the number of times the DOO algorithm went through Line 5 strictly before that time, and denote by $n_\varepsilon$ the total number of evaluations of $f$ strictly before that same time. We have

$$n_\varepsilon \leq 1 + K I_\varepsilon .$$

Furthermore, after $n_\varepsilon$ evaluations of $f$, we have, by definitions of the recommendation $\boldsymbol{x}_{n_\varepsilon}^\star$ and $n_\varepsilon$,

$$f(\boldsymbol{x}_{n_\varepsilon}^\star) = \max_{\boldsymbol{x} \in \{\boldsymbol{x}_1, \ldots, \boldsymbol{x}_{n_\varepsilon}\}} f(\boldsymbol{x}) \geq f(\boldsymbol{x}_{h^\star, i^\star}) \geq f(\boldsymbol{x}^\star) - \varepsilon .$$

This inequality entails that the non-certified sample complexity of nc.DOO is bounded by $n_\varepsilon$ and thus

$$\zeta(\text{nc.DOO}, f, \varepsilon) \leq 1 + K I_\varepsilon. \tag{18}$$

We now bound $I_\varepsilon$ from above, and assume without loss of generality that $I_\varepsilon \geq 1$. Consider now the sequence $(h_1^\star, i_1^\star), \ldots, (h_{I_\varepsilon}^\star, i_{I_\varepsilon}^\star)$ corresponding to the first $I_\varepsilon$ times the DOO algorithm nc.DOO went through Line 5. Let $\mathcal{E}_\varepsilon$ be the corresponding finite set $\{\boldsymbol{x}_{h_1^\star, i_1^\star}, \ldots, \boldsymbol{x}_{h_{I_\varepsilon}^\star, i_{I_\varepsilon}^\star}\}$ (a leaf can never be selected twice). By definition of $I_\varepsilon$, we have $\mathcal{E}_\varepsilon \subseteq \mathcal{X}_{(\varepsilon, \varepsilon_0]}$. Since $\varepsilon = \varepsilon_{m_\varepsilon} \leq \varepsilon_{m_\varepsilon - 1} \leq \ldots \leq \varepsilon_0$, we have $\mathcal{E}_\varepsilon \subseteq \bigcup_{i=1}^{m_\varepsilon} \mathcal{X}_{(\varepsilon_i, \, \varepsilon_{i-1}]}$, so that the cardinality $I_\varepsilon$ of $\mathcal{E}_\varepsilon$ satisfies

$$I_\varepsilon = \text{card}(\mathcal{E}_\varepsilon) \leq \sum_{i=1}^{m_\varepsilon} \text{card}\left(\mathcal{E}_\varepsilon \cap \mathcal{X}_{(\varepsilon_i, \, \varepsilon_{i-1}]}\right) . \tag{19}$$

Let $N_{\varepsilon, i}$ be the cardinality of $\mathcal{E}_\varepsilon \cap \mathcal{X}_{(\varepsilon_i, \, \varepsilon_{i-1}]}$. The same arguments as from (13) to (15) in the proof of Proposition 1 yield

$$N_{\varepsilon, i} \leq \left(\mathbf{1}_{\nu/R \geq 1} + \mathbf{1}_{\nu/R < 1}\left(\frac{4R}{\nu}\right)^d\right) \mathcal{N}\left(\mathcal{X}_{(\varepsilon_i, \, \varepsilon_{i-1}]}, \frac{\varepsilon_i}{L}\right). \tag{20}$$

Combining the last inequality with (18) and (19) concludes the proof. □

**Remark 4.** *The analysis of the DOO algorithm in Munos (2011, Theorem 1) (non-certified version) yields a bound on the non-certified sample complexity (6) than can be expressed in the form $1 + C \sum_{k=1}^{m_\varepsilon} \mathcal{N}\left(\mathcal{X}_{\varepsilon_{k-1}}, \frac{\varepsilon_k}{L}\right)$, with a constant $C$. The corresponding proof relies on two main arguments. First, when a cell of the form $(h^\star, i^\star)$, $i^\star \in \{0, \ldots, K^{h^\star} - 1\}$, is selected in Line 5 of Algorithm 1, then the corresponding cell representative $\boldsymbol{x}_{i^\star, h^\star}$ is $LR\delta^{h^\star}$-optimal (we also use this argument). Second, as a consequence, for a given fixed value of $h^\star$, for the sequence of values of $i^\star$ that are selected in Line 5 of Algorithm 1, the corresponding cell representatives $\boldsymbol{x}_{h^\star, i^\star}$ form a packing of $\mathcal{X}_{LR\delta^{h^\star}}$.*

*Our slight refinement in the proof of Proposition 5 stems from the observation that using a packing of $\mathcal{X}_{LR\delta^{h^\star}}$ yields a suboptimal analysis, since the cell representatives $\boldsymbol{x}_{h^\star, i^\star}$ can be much better than $LR\delta^{h^\star}$-optimal. Hence, we proceed differently from Munos (2011), by first partitioning all the selected cell representatives (in Line 5 of Algorithm 1) according to their level of optimality as in (19) and then by exhibiting packings of the different layers of input points $\mathcal{X}_{(\varepsilon, \varepsilon_{m_\varepsilon - 1}]}, \mathcal{X}_{(\varepsilon_{m_\varepsilon - 1}, \varepsilon_{m_\varepsilon - 2}]}, \ldots, \mathcal{X}_{(\varepsilon_1, \varepsilon_0]}$. In a word, we partition the values of $f$ instead of partitioning the input space when counting the representatives selected at all levels.*

**Remark 5.** *In the Introduction, below Eq. (6), we mentioned the inherent difference between the sample complexity $\sigma(A, f, \varepsilon)$ in the certified setting and the more classical sample complexity $\zeta(A, f, \varepsilon)$ in the non-certified setting. We can now make our statements more formal.*

*Our paper shows that in the certified setting, the sample complexity $\sigma(A, f, \varepsilon)$ of an optimal algorithm $A$ (e.g., $A = \text{c.DOO}$) is characterized by the quantity*

$$S_\text{C}(f, \varepsilon) := \mathcal{N}\left(\mathcal{X}_\varepsilon, \frac{\varepsilon}{L}\right) + \sum_{k=1}^{m_\varepsilon} \mathcal{N}\left(\mathcal{X}_{(\varepsilon_k, \varepsilon_{k-1}]}, \frac{\varepsilon_k}{L}\right) .$$

*In contrast, the previous proposition shows that in the non-certified setting the sample complexity $\zeta(\text{nc.DOO}, f, \varepsilon)$ of the nc.DOO algorithm is upper bounded (up to constants) by*

$$S_\text{NC}(f, \varepsilon) := \sum_{k=1}^{m_\varepsilon} \mathcal{N}\left(\mathcal{X}_{(\varepsilon_k, \varepsilon_{k-1}]}, \frac{\varepsilon_k}{L}\right) .$$

*The two expressions look remarkably alike but are subtly very different. In fact, the latter depends only on the "size" (i.e., the packing numbers) of suboptimal points. The former has an additional term measuring the size of near-optimal points. Now, note that the flatter a function is, the fewer suboptimal points there are. This implies that the sum $\sum_{k=1}^{m_\varepsilon} \mathcal{N}\left(\mathcal{X}_{(\varepsilon_k, \varepsilon_{k-1}]}, \frac{\varepsilon_k}{L}\right)$ becomes very small (hence, so does $S_{\mathrm{NC}}(f, \varepsilon)$), but in turn, the set of near-optimal points $\mathcal{X}_\varepsilon$ becomes large (hence, so does $S_{\mathrm{C}}(f, \varepsilon)$). For instance, in the extreme case of constant functions $f$, we have $S_{\mathrm{NC}}(f, \varepsilon) = 0$ but $S_{\mathrm{C}}(f, \varepsilon) \approx (L/\varepsilon)^d$. This fleshes out the fundamental difference between certified and non-certified optimization, giving formal evidence to the intuition that the more "constant" a function is, the easier it is to recommend an $\varepsilon$-optimal point, but the harder it is to certify that such recommendation is actually a good recommendation.*