# OpenReview forum: "Instance-Dependent Bounds for Zeroth-order Lipschitz Optimization with Error Certificates"
_NeurIPS.cc/2021/Conference — NeurIPS 2021 Poster_

### Official Review · Reviewer_TJAZ · 2021-07-05

**Rating:** 6
**Confidence:** 2

**Summary:**

This work gives nearly tight upper/lower bounds for the sample complexity of optimizing any lipschitz function using zeroth-order oracles with certification. They prove the sample complexity is proportional to the integral of  $1/(\max(f)-f(x)+\epsilon)^d$ up to a $\log \epsilon$ factor, which is instance-dependent.

**Limitations And Societal Impact:**

Yes

**Main Review:**

Originality: This paper considers the version of zeroth-order lipschitz optimization with certification. Considering certification is novel, but its motivation needs further justification. For example, is the goal of certification only to certify the $\epsilon$-optimal point, or to certify $every$ point generated by the algorithm? If we only want to provide certification for the $\epsilon$-optimal point, then bandit optimization algorithms can do as well by simply picking the best point played in hindsight after playing enough rounds. The goal to certify the output of the algorithm generated in every round, in the contrary, seems extravagant since we don't care about certification when the output is far away from optimal.

Quality: It's not too surprising that one can get instance-dependent bounds, since certification essentially requires 'exploring every corner of $X$' at a high level. My main concern is on the constant $c$ and its dependence on $d$. Roughly speaking, $c$ is the product of two terms, where the first is $L^d/v_1$ and the second is $(1-Lip(f)/L)^d$. The second term is exponentially small (in $d$), which is problematic (it hurts the tightness of the upper bound) even if $d$ is seen as a constant. The first term is either exponentially large or exponentially small in $d$, depending on whether $L$ is above a constant threshold. This seems unnatural as one can easily scale the funtion to be $l$-lipschitz for arbitrary $l$, and finding an $\epsilon l/L$-optimal point will suffice. Moreover, $\gamma$ isn't completely independent of $d$ either, because the value of $\gamma$ would be not mild at all for large $d$ whereas the same value might be mild for low dimension.

Clarity: This paper is overall well-written, except that the end of Section 4 ($L=Lip(f)$ case) should be written in a separate section with more details. Adding proof sketches instead of putting technical proofs in the maintext would improve readability.

Significance: This work solves an existing open problem. Though whether the proposed algorithm could be useful in practice is unclear, it's of importance from a theoretical perspective.


**Time Spent Reviewing:**

12

---

> ### Author Response · Authors · 2021-08-10
> **Response to TJAZ**
>
> We thank the reviewer for the time spent reading our paper and for sharing their comments.
> We will update the revised version in light of your feedback.
>
> ---
>
> *Clarifications on the setting*
>
> Let us summarize here the setting of certified optimization. At each round $n$, a certified algorithm does three things: 1) it picks a query point $x_n$, 2) it observes $f(x_n)$, 3) it outputs a recommendation $x_n^\star$ together with an error certificate $\xi_n$. The error certificate relates to the *recommendation*, not the *queries*. We agree that it would be quite extravagant to try certifying the queries. Note that (although in the certified DOO algorithm this happens) in general, $x_n^\star$ does not even need to belong to $x_1,\ldots,x_n$ so that the two things should not be confused.
> We want the error certificate $\xi_n$ to be larger than $\max(f) - f(x^\star_n)$ (not $f(x_n)$).
> We will make sure that this is more emphasized in the revised version.
>
> *Classic bandit optimization vs. certified optimization*
>
> We assume the reviewer refers to classic *non-certified* (bandit) optimization here.
> Let us stress the differences between the theoretical $f$-dependent guarantees in non-certified vs. certified optimization. Imagine the two following theorems are proved.
>
> Theorem A (statement for non-certified optimization): for any function $f$ and number of evaluations $n$, the recommendation $x^\star_n$ returned by a non-certified algorithm satisfies $\max(f)-f(x^\star_n) \le g(f,n)$, for some guarantee function $g$.
>
> Theorem B (statement for certified optimization): for any function $f$ and number of evaluations $n$, the recommendation $x^\star_n$ and certificate $\xi_n$ returned by a certified algorithm satisfy $\max(f)-f(x^\star_n) \le \xi_n \le h(f,n) $, for some guarantee function $h$.
>
> Imagine one tries to apply Theorem A in real life. They run the non-certified algorithm for $n$ rounds, produce a recommendation $x^\star_n$, and then wonder how good the $x^\star_n$ is. There is no way of knowing because the guarantees depend on the *unknown* function $f$. Essentially, the only way to have some idea is upper bounding $g(f,n)$ with the (way) looser $\sup_{f'} g(f', n)$.
>
> On the other hand, Theorem B knows that the approximation is demonstrably $\xi_n$-optimal or better.
>
> For this reason, certified optimization is much more powerful than classic non-certified optimization. Of course, this comes at a cost, that the convergence guarantees are slower.
>
> From a theoretical perspective, the certified setting makes it also possible to consider instance-dependent lower bounds (Theorem 2) and thus to obtain tight instance-dependent bounds (Theorem 3) which is a novelty of certified optimization (the lower bounds would be trivially $=1$ in the non-certified setting). Furthermore, even upper bounding the sample complexity in the certified setting requires novelties compared to the non-certified setting. The reason is, intuitively, that, when the optimization error happens to be smaller than $\varepsilon$,  a certified algorithm typically still has to query more points to be able to certify this. For additional details, we refer the reviewer to lines 131-141 and to the comparison between the proofs of Proposition 5 (non-certified DOO) and of Proposition 1 (certified DOO).
>
> We hope this clarifies things. We will highlight this more in the paper.
>
> *Constants depending exponentially on $d$.*
>
> In this paper, we admittedly did not focus too much on finding the tightest possible multiplicative constants. As such, one could certainly expect to improve them even by simply applying more accurately our very same arguments. However, we believe that eliminating the exponential mismatch of upper and lower bounds altogether would be extremely challenging. This problem is common when working in high-dimensional spaces: any looseness in the analysis propagates in $d$ dimensions, giving, in turn, an exponential deterioration of the constants. These considerations do not at all dismiss the reviewer's comments. On the contrary, this is the next big open problem in certified optimization! We will make sure to state more clearly the dependencies within the multiplicative constants.
>
> Furthermore, it may seem surprising in Theorem 1 to have $c = 1/v_{1/L} = L^d / v_1$ and $C = 1 / (\gamma v_{1/128 L} ) = 128^d L^d / \gamma  v_1$, where in both cases the threshold $L=1$  has a special role.
> The explanation is with the expression of $S_{\mathrm{C}}(f, \varepsilon)$,
> $$S_\mathrm{C}(f,\varepsilon) =
> \mathcal{N} \left( \mathcal X_{\varepsilon}, \frac{\varepsilon}{L} \right) + \sum_{k=1}^{m_{\varepsilon}} \mathcal{N} \left( \mathcal X_{(\varepsilon_k, \varepsilon_{k-1}]}, \frac{\varepsilon_k}{L} \right)$$
> which scales like $L^{d}$, i.e., as a function of $L$ and $d$ (when $L$ is large, the packing number is larger because it is defined with a smaller scale). In Theorem 1, the integral bound does not depend on $L$, which is why it is multiplied by the constants $c$ and $C$: to compensate the fact that $S_\mathrm{C}(f,\varepsilon)$ does scale with $L^d$.
>
> Finally, we agree that when $d$ is large, one may need to take a smaller $\gamma$ to satisfy Assumption 4. For instance, when $\mathcal{X}$ is a hypercube, we would need $\gamma = 2^{-d}$. We will mention this in the revised version.
>
> *Give a separate section to the case $L=\mathrm{Lip}(f)$*
>
> Good idea! We will move the end of Section 4 ($L = \mathrm{Lip}(f)$ case) to a separate section.
> This choice will also facilitate adding some of the comments that came up from the other reviewers.
> Thanks for the suggestion!
>
>
> *Is the proposed algorithm useful in practice?*
>
> The certified DOO algorithm, that we study theoretically here, can indeed be useful in practice. First, certified optimization is more informative and trustworthy than non-certified one (see our response above as well as the motivation in the first paragraph of the introduction). Furthermore, the computational cost of certified DOO is significantly smaller than that of other existing certified algorithms (e.g., the certified Piyavskii-Shubert algorithm, as we mention briefly at the end of page 3). This makes certified DOO more useful in practice. Finally, non-certified algorithms sharing with certified DOO the idea of hierarchical partitioning of the input space tend to be used in practice, see for instance the survey: R. Munos, "From Bandits to Monte-Carlo Tree Search: The Optimistic Principle Applied to Optimization and Planning", Foundations and Trends in Machine Learning Vol. 7, No. 1 (2014) 1–129.
>
> ---
>
> We hope we were able to provide all the relevant clarifications and give satisfactory answers to all questions. If not, we are happy to reply to any follow-up questions during the discussion period. If so, we would be happy if the reviewer could consider raising their score.

---

> > ### Comment · Reviewer_TJAZ · 2021-08-18
> > **Answer to authors**
> >
> > Thank you for your detailed response, especially clarification on the setting. I will keep the current score though, because I feel that there is still a lot to do before achieving a satisfactory dimension dependency.

---

### Official Review · Reviewer_6ukK · 2021-07-06

**Rating:** 7
**Confidence:** 4

**Summary:**

The paper studies the complexity of certifiable zeroth-order optimization of Lipschitz functions. Formally, given oracle access to evaluations of a $L$-Lipschitz function $f$ over a known compact domain $\mathcal{X}$ and an error parameter, $\epsilon > 0$, the goal is to find a point $\hat{x}$ satisfying $f (\hat{x}) \geq \max_x f (x) - \epsilon$ making as few oracle queries as possible. The paper characterizes the optimal oracle complexity of the problem in an instance dependent sense where the number of queries made to the oracle depend on the behavior of the particular function. Concretely, they show that the number of queries that necessarily must be made to the oracle is $\int_{\mathcal{X}} d \mathbf{x} / (\max(f) - f(\mathbf{x})  + \epsilon)^d$ up to constant factors depending (exponentially) on the dimension.

**Limitations And Societal Impact:**

Yes

**Main Review:**

The paper studies the complexity of certifiable zeroth-order optimization of Lipschitz functions. Formally, given oracle access to evaluations of a $L$-Lipschitz function $f$ over a known compact domain $\mathcal{X}$ and an error parameter, $\epsilon > 0$, the goal is to find a point $\hat{x}$ satisfying $f (\hat{x}) \geq \max_x f (x) - \epsilon$ making as few oracle queries as possible. The paper characterizes the optimal oracle complexity of the problem in an instance dependent sense where the number of queries made to the oracle depend on the behavior of the particular function. Concretely, they show that the number of queries that necessarily must be made to the oracle is $\int_{\mathcal{X}} d \mathbf{x} / (\max(f) - f(\mathbf{x})  + \epsilon)^d$ up to constant factors depending (exponentially) on the dimension. The right notion of complexity in the instance dependent setting is a little tricky as the optimal algorithm in the instance dependent setting would be one that simply outputs the optimizer of the particular function in question. However, if one additionally imposes the requirement of certification which is essentially that the algorithm must output a correct answer for \emph{any} Lipschitz function over $\mathcal{X}$, one obtains nontrivial bounds.

On a technical level, there seem to be two main technical contributions in the paper. Firstly, noting that a modification of the Deterministic Optimistic Optimization algorithm by Munos (2011) obtains a sample complexity of:
$$
\mathcal{N} (\mathcal{X}\_\epsilon, \epsilon / L) + \sum\_{k = 1}^{m\_\epsilon} \mathcal{N} (\mathcal{X}\_{(\epsilon\_k, \epsilon\_{k - 1}]}, \epsilon\_k / L)
$$
where $m_\epsilon = \log \epsilon_0 / \epsilon$, $\epsilon_k = \epsilon_0 2^{-k}$,  $\epsilon_0 = \max_{x,y \in \mathcal{X}} L \lVert x - y \rVert$, $\mathcal{X}\_\epsilon = \{x \in \mathcal{X}: f(x) \geq \max (f) - \epsilon \}$, $\mathcal{X}\_{(a, b]} = \{x \in \mathcal{X}: a < \max(f) - f(x) \leq b\}$ and $\mathcal{N} (\mathcal{A}, \gamma)$ denotes a $\gamma$-packing number of $\mathcal{A}$, they show that under reasonable assumptions on $\mathcal{X}$, the integral is a dimension-dependent constant factor approximation of the integral characterizing the optimal upper and lower bounds. Secondly, they show any certified algorithm must incur an oracle complexity of:
\begin{equation*}
    c \int_{\mathcal{X}} \frac{d \mathbf{x}}{(\max(f) - f(\mathbf{x}) + \epsilon)^d} \text{ where } c = c' \frac{(1 - \mathrm{Lip} (f) / L)^d}{1 + m_\epsilon}
\end{equation*}
providing some justification on why $1 - \mathrm{Lip} (f) / L$ factor in the lower bound may be inherent. In proving the lower bound, they show that any algorithm making fewer than the number characterized by the lower bound for some function $f$ must make an error on a related function $f'$ constructed by perturbing $f$ using a precisely constructed bump function around a close-to-optimal point of $f$. The construction itself is fairly intuitive but yields near-optimal results.

Overall, the paper makes progress on a natural question relating to a fundamental problem that is of interest to the NeurIPS community. The proofs themselves are quite natural and intuitive though one small caveat is that the assumption on $\mathcal{X}$ (Assumption 4) while seemingly reasonable is a little difficult to reason about. Additional exposition on the types of sets for which it may (or may not) hold would be quite helpful.

**************************** POST REBUTTAL ADJUSTMENTS ****************************

After reviewing the authors response, I am convinced of the technical quality of the results in the paper and the assumptions underlying them. I will retain my recommendation to accept the paper.

**Time Spent Reviewing:**

5

---

> ### Author Response · Authors · 2021-08-10
> **Response to Reviewer 6ukK**
>
> We thank the reviewer for the time spent reading our paper and for sharing their comments.
> We will update the revised version in light of your feedback.
>
> ---
>
> You could see Assumption 4 as a relaxation of convexity, in the sense that all convex sets with a non-empty interior satisfy it.
> To see it, consider a fixed ball $B(x_0,r_0)$ in $\mathcal{X}$ with $r_0>0$. To each $x \in \mathcal{X}$ we can associate the set $C_x$ of all the segments which start at $x$ and end in $B(x_0,r_0)$ which is included in $\mathcal{X}$. Then, we can show that the volume of $B(x,r) \cap C_x$ is lower bounded by a constant  times $r^{-d}$ as $r \to 0$ (using that $\mathcal{X}$ is bounded).
> As a consequence, Assumption 4 also holds for finite unions of convex sets with non-empty interiors.
>
> More broadly, Assumption 4 gives a quantitative guarantee that $\mathcal X$ is "really" $d$-dimensional: none of its connected components can be embedded in $\mathbb R^{d'}$ (with $d'<d$) nor have cusps.
>
> The role of the assumption is to make $d$-dimensional integration on $\mathcal{X}$ meaningful, in particular excluding cases where $\mathcal{X}$ is a manifold of dimension strictly less than $d$ embedded in $\mathbb{R}^d$.
> It is easy to show that we cannot remove it altogether (not without replacing it with some similar condition).
> Indeed, if, for example, $\mathcal{X}$ were a $d-1$-dimensional manifold embedded in $\mathbb{R}^d$, then the integral bound would be zero, but the bound based on packing numbers would still be non-zero, violating Theorem 1.
>
> We will add a little more intuition on Assumption 4 in the revised version.

---

### Official Review · Reviewer_QEpC · 2021-07-16

**Rating:** 7
**Confidence:** 4

**Summary:**

This paper studied the problem of zeroth-order Lipschitz optimization, and mainly showed that under mild geometric assumptions and a noise-free zeroth-order oracle, the sample complexity of finding the \varepsilon-optimal point for a function f is of the order $\int dx/(f(x^\star) - f(x) + \varepsilon)^d$ (up to logarithmic factors). More specifically, the main contributions are as follows:

1. The authors proposed a certified DOO algorithm which could provide an error certification at each step, and achieves a sample complexity provided by the sum of packing numbers S_C(f,\varepsilon), a quantity already proposed in Bouttier et al. (2020).

2. Under a mild geometric assumption, the quantity S_C(f,\varepsilon) is bounded by the integral from both above and below, thereby proving the upper bound.

3. For the instance-dependent lower bound, the authors used a simple perturbation idea to reduce to a proper packing number: if the packing number of some shell is too large, then there must be points not queried by the algorithm, and a local perturbation of the functions around the non-queried points made the certification break down.

**Main Review:**

Overall this is a nice paper and I enjoyed reading it. This paper provides a complete characterization for the instance-dependent complexity of zeroth-order Lipschitz optimization, solving a long-standing conjecture for over 30 years.

If I have to list some weaknesses, perhaps the main weakness is that the current paper may owe a lot to Bouttier et al. (2020) which obtained the crucial quantity S_C(f,\varepsilon). Based on this quantity the remaining steps are actually not so hard, including its relationship to the integral, and the lower bound argument. Therefore, I suggest the authors to discuss more on the importance of error certification, as well as the main novelty/difference/challenge in the current DOO algorithm compared with the previous ones. This would make the contribution of the current paper more convincing.

Some additional comments:

1. Perhaps it would be nice to add that error certification makes the instance-dependent complexity meaningful. Without error certification, the complexity lower bound must be in a local minimax sense instead of strictly pointwise.

2. In the lower bound, does any result change if the points could be queried randomly? The pigeonhole principle always becomes a little bit shaky to apply when things are not deterministic. Some discussions would be appreciated.

3. Also, if the oracle is actually noisy, do the authors have any intuition/evidence whether the sample complexity is roughly the same, or changes dramatically? Some discussions would also be appreciated.

4. Line 66-68: check for language issues.

Post-rebuttal feedback:

Thanks to the authors for the detailed response. Please add the discussions on the random query model and noisy oracles if space permits. I am happy to keep my score and recommend acceptance.

**Time Spent Reviewing:**

3

---

> ### Author Response · Authors · 2021-08-10
> **Response to QEpC**
>
> We thank the reviewer for the time spent reading our paper, for their positive feedback, and their suggestions.
> We will update the revised version in light of your comments.
>
> ---
>
> *Suggestion to discuss more on the importance of error certification.*
>
> We agree with that. We will add a few sentences after Eq. (2) in the introduction to explain why certification can really matter in practice.
>
> Furthermore, certified optimization yields different theoretical upper bounds than non-certified optimization, and we believe it is important to understand and characterize this difference. We have some discussions about this after (6) and in Remark 5 (supplementary material).
>
> Finally, as you write in your additional comment 1, considering certified-optimization allows for meaningful instance-dependent lower bounds. We will add a sentence to the last paragraph of Section 1.3. Thanks for the suggestion!
>
> *Suggestion to discuss more on the main novelty/difference/challenge in the current DOO algorithm compared with the previous ones.*
>
> Thank you for the suggestion! We see several points that we can highlight slightly more in the paper.
>
> - First, the certified DOO algorithm is (by definition) more informative than the previous non-certified versions. This comes at a sample complexity cost (more evaluations of $f$ are required, as detailed in lines 131-141 and in Section E), and determining this cost is a challenging important question.
>
> - Second, compared to Piyavskii-Shubert (Bouttier et al. 2020), our certified DOO has a much smaller computational cost (as we mention at the end of page 3 and in Remark 1). This however comes at a theoretical (challenging) cost, as explained below.
>
> The Piyavskii-Shubert algorithm relies on maximizing the maximum possible value of $f$. This computationally limiting feature actually makes it easier theoretically to provide an optimization certificate and to analyze the sample complexity. With certified DOO, the error certificate is constrained by the tree structure (line 13 of Algorithm 1), which makes the optimal analysis of the sample complexity more challenging.
>
> - Third, the existing analysis of DOO in the non-certified setting used arguments that are not as tight as ours (see Remark 2 in Section B). We had to slightly improve these arguments to get the bound of Proposition 1. We will make this clearer in the introduction.
>
> *Other comments.*
>
> 1. Great suggestion! We will make sure to mention it in the revised version.
>
> 2. Good question! Actually, for randomized algorithms, it is not only the analysis (e.g., the pigeonhole principle) that becomes more subtle, but the very meaning of a lower bound for randomized algorithms is also subtle. We see at least three different questions.
>
> - Assume we define a randomized algorithm $A$ to be certified when, with probability one (with respect to the random seed of the algorithm), for all $L$-Lipschitz functions $f$, for all $n \geq 1$, $\max(f) - f(x_n^\star) \leq \xi_n$. Assume also that the sample complexity $\sigma(A,f,\varepsilon)$ is defined as in eq. (2), so that it is now a random variable. Then, we can apply Theorem 2 conditionally on the random seed of the algorithm to show that, with probability one, for all $f$, the sample complexity is lower bounded as in Theorem 2. (This is in particular also true in expectation or for any quantile of $\sigma(A,f,\varepsilon)$.) The only subtlety here is that the two balls that $A$ has not visited in the proof of Theorem 2 may depend on the random seed of $A$, which is fine since $A$ is certified for every perturbation of $f$.
>
> - Assume now that, by a certified randomized algorithm $A$, we mean: for all $L$-Lipschitz functions $f$, with probability one, for all $n \geq 1$, $\max(f) - f(x_n^\star) \leq \xi_n$. (We just made the certification constraint less stringent by inverting "for all $f$" and "with probability one".) Now, since the latter probability-one event may depend on $f$ (or on any other perturbation function that we could consider), and since the set of $L$-Lipschitz functions is uncountable, Theorem 2 cannot be applied immediately. However, for any fixed $f$, the set of all possible perturbation functions that we consider in the proof of Theorem 2 is finite, so that, with probability one, the algorithm is simultaneously certified for $f$ and all these possible perturbation functions. Therefore, for any $f$, with probability one, the lower bound of Theorem 2 applies.
>
> - Another related question is if we allow randomized algorithms to be only certified with high probability, instead of probability one. We do not know yet if that enables to considerably lower the sample complexity quantiles or not. We will add this interesting question to the open problems section.
>
> 3. Another good question! We have some intuition that with zero-mean subgaussian (or even heavier-tailed) i.i.d. perturbations the upper bound on the complexity should look something like $\int_{\mathcal X} \frac{\mathrm d x}{(\max(f) - f(x) + \varepsilon)^{d+2}}$ with an extra $+2$ in the exponent of the denominator compared to the deterministic case.
> As in our paper, we first prove the result with packing numbers. The idea is to get an expression like $S_{\mathrm C}(f,\varepsilon)$ but where each packing number $\mathcal N (\mathcal X_{(\varepsilon_k, \varepsilon_{k-1}]},\varepsilon_k/L)$ (resp., $\mathcal N (\mathcal X_{\varepsilon},\varepsilon/L)$) is multiplied by $\approx 1/\varepsilon_k^2$ (resp., $1/\varepsilon^2$) because to see an $\varepsilon_k$ (resp., $\varepsilon$)-big gap through the variance, the learner has to draw roughly $\approx 1/\varepsilon_k^2$ (resp., $1/\varepsilon^2$) samples. Note that some adaptivity argument is required here because when the learner queries a point $x\in \mathcal X$, they do not know to which layer $\mathcal X_{(\varepsilon_k, \varepsilon_{k-1}]}$ (or $\mathcal X_{\varepsilon}$) $x$ belongs. Once this new packing-number bound is established, one could get the integral proceeding as in our paper.
>
> 4. Thanks for spotting it!

---

### Official Review · Reviewer_FGKi · 2021-07-22

**Rating:** 7
**Confidence:** 3

**Summary:**

  In this paper, the authors consider the problem of finding global solutions
  to a possibly non-convex optimization problem under possibly non-convex
  constraints using only access to function values of the objective function.
  In particular, they focus on certified algorithms, i.e., algorithm that
  together with a solution they also output certificate of the optimality of the
  solution.

  Their main result is an instance optimal bound for any Lipschitz continuous
  functions. In particular if f is a Lipschitz continuous function with unknown
  Lipschitz constant that is less than L, then the number of zero-order queries
  needed to find an $\epsilon$-approximate global optimum together with a
  certificate of optimality is

  $\sigma = \int_{\mathcal{X}} \frac{1}{(f(x^*) - f(x) + \epsilon)^d} dx$

  where $f(x^*)$ is the value of the global optimum and d is the dimensionality
  of the problem. The authors provide both an algorithm and a lower bound that
  is equal to $\sigma$ up to dimension dependent constants. Surprisingly the fact
  that the Lipschitz constant is not exactly known but only upper bounded is
  crucial for the proof of the lower bound.

**Limitations And Societal Impact:**

Yes

**Main Review:**

Strengths
  ====================

  - The paper resolves an interesting open question from 1991.

  - The problem of finding global optima with zero-order methods is very
  important in many areas, e.g., in hyperparameter tuning.

  - The techniques developed to prove the results are very interesting.

  - The use of the notion of certified algorithms is a very clever way to be
  able to accurate prove lower bounds for this problem.


  Weaknesses - Comments
  ========================

  1. It would be nice to have some better understanding of the expression of
  \sigma. For example, when the domain is [0, 1]^d, a trivial brute force
  algorithm would need (L/\epsilon)^d steps to get a certified
  \epsilon-approximately optimal solution. How is this bound compared to the
  per-instance optimal bound? It is clear that the per-instance bound is
  smaller but is which cases it is significantly smaller? For example if L is
  O(1) are the two bounds of the same order.

  2. The term "constant" in many lemmas and theorems is misleading. In many
  cases the constants C, c, or c' dependent at least exponentially to the
  dimension d of the problem which is a very important parameter. I understand
  that these constants do not dependent on f and \epsilon but d is an important
  parameter as well and the authors should be more careful when they use the
  term constant that has some dependence on d.

  3. The use of the term "sample complexity" is a bit misleading a believe a
  better wording would be "query complexity".

  4. In lines 244-250 it is mentioned that Assumption 4 is satisfied for many
  natural domain sets. Is it satisfied in general for convex sets X or there is
  a counter-example?

  5. In the proof of Theorem 2 in line 302 the notation K is confusing because K
  has been used so far for the expansion of the tree in the algorithm.

  6. The fact that the exact knowledge of the Lipschitzness is fundamental for
  the lower bound is a bit confusing. It is possible that the exact knowledge
  of the Lipschitz constant can be used to get an improved algorithm for any
  function f?


  Summary of Recommendation
  ======================================
  The results provided in this paper seem to be important and significant and
  for this reason I recommend acceptance.


  Post-Rebuttal
  ======================================
  After reading the authors response and the other reviews I keep my score.


**Time Spent Reviewing:**

8

---

> ### Author Response · Authors · 2021-08-10
> **Response to Reviewer FGKi**
>
> We thank the reviewer for the time spent reading our paper and for sharing their comments.
> We will update the revised version in light of your feedback.
>
> ---
>
> *On the exact knowledge of the best Lipschitz constant.*
>
> We agree that it is surprising and potentially confusing that the lower bound $S_C(f, \varepsilon)$ is not tight in dimension $d \geq 2$ when $\mathrm{Lip}(f) = L $ (i.e., when the best Lipschitz constant is known exactly). While this was surprising to us at first, we now have two intuitive explanations for the particular role of the assumption $\mathrm{Lip}(f) = L $.
>
> First, when $\mathrm{Lip}(f) = L $, the sample complexity can be surprisingly smaller for some functions.
> This is because there can be pairs of points at which the respective values of $f$ are maximally distant (slope $L$), which can help the algorithm.
> This is what happens in the proof of Proposition 2, where two such points (say $u$ and $v$) enable to have an error certificate that is exactly zero. Indeed, the distance between $u$ and any point of the domain is smaller or equal to the distance between $u$ and $v$, and thus $v$ is guaranteed to be a maximizer from the Lipschitz property and from $f(v) = f(u) + L ||u-v||$.
> A simpler example that brings some more intuition in dimension $1$ along these lines is when the domain is $[0,1]$, $L=1$, and we observe $f(0) = 0$ and $f(1) = 1$. Then we know that the $1$-Lipschitz function is the identity on $[0,1]$, from only these two observations.
>
> Second, when $\mathrm{Lip}(f) = L $, $S_C(f , \epsilon)$ is not a tight lower bound in dimension $d \geq 2$ because even when it is large, it may be impossible to construct $L$-Lipschitz functions that coincide with $f$ on the sample/query points and that "trap" the algorithm when $n$ is much smaller than $S_C(f , \epsilon)$. Indeed, in the proof of Theorem 2, we start from a Lipschitz function $f$ with $\mathrm{Lip}(f) < L $ and thus, whenever there is an empty ball, we can increase or decrease the values of the function by adding $\pm 1$ times a  "pyramid" with slope $L - \mathrm{Lip}(f)$.  When $f$ is such that $\mathrm{Lip}(f) = L $, then it is impossible to add such pyramids and thus to "trap" the algorithm in the same way. Note also that, surprisingly, the case $d=1$ behaves differently (proof of Proposition 3). When $d=1$, $\mathrm{Lip}(f) = L$ and $S_C(f , \epsilon)$ is large, we can still "trap" the algorithm by modifying $f$ in a way that is specific to the geometry of the real line.
>
> These intuitive discussions above also appear in our answers to other reviewers, who also had comments related to the case $\mathrm{Lip}(f) = L$. We will add this discussion to the paper to give more insight on the case $\mathrm{Lip}(f) = L$.
>
> We are grateful to you and the other reviewers for bringing this point to our attention.
>
> 1. Here are a few examples that show how the optimal sample complexity is (in general) much better than the brute force $(\varepsilon / L)^d$.
> For the sake of simplicity, take $L=1$.
> First, in dimension $d = 1$, if the domain is $\mathcal{X} = [0,1]$ and if $f(x) = 1-x$, then one can show that the integral upper bound is of order $\ln (1/\varepsilon)$ as $\varepsilon \to 0$, which is much smaller than $1 / \varepsilon$. Similarly, in general dimension $d$, if the function $f$ satisfies $f(x) = f(x^\star) - ||x - x^\star||$ for some maximizer $x^\star$, then the integral bound is upper bounded by a constant times $\int_{B(0,r_0)} (\varepsilon + ||x||)^{-d} \mathrm{d}x $, with $B( 0 , r_0)$ being a ball of fixed radius $r_0$. By a polar change of variables followed by $d-1$ integration by parts, this integral is again of order $\ln(1/\varepsilon)$. Finally, in dimension $d=4$, if the function $f$ has a unique maximizer $x^\star$ and satisfies $f(x) \leq f(x^\star) - ||x - x^\star||^2$ around $x^\star$, then the integral bound is upper bounded by a constant times $\int_{B(0,r_0)} (\varepsilon + ||x||^2)^{-d} \mathrm{d}x $. By a polar change of variables followed by an integration by parts, the integral is of order $1/\varepsilon^{2}$ (again smaller than $1/\varepsilon^{4}$ for the brute force algorithm).
>
>
> 2. As suggested, we will mention explicitly that the constants (e.g. $c'$ and $C$ in Theorem 3) depend exponentially on $d$. We thank you and the other reviewers that noticed it for catching that we were not always consistent in giving out explicit constants (which is what we usually prefer doing).
> We do agree that while being constant with respect to $f$ and $\varepsilon$, $d$ is still very much a notable parameter.
>
> 3. Fair point!
> We will remark that what is usually referred to as *sample* complexity could also (/should instead) be called *query complexity*.
>
> 4. Great question! You could indeed see Assumption 4 as a sort of relaxation of convexity, in the sense that all convex sets with a non-empty interior satisfy it.
> To see it, consider a fixed ball $B(x_0,r_0)$ in $\mathcal{X}$ with $r_0>0$. To each $x \in \mathcal{X}$ we can associate the set $C_x$ of all the segments which start at $x$ and end in $B(x_0,r_0)$ which is included in $\mathcal{X}$. Then, we can show that the volume of $B(x,r) \cap C_x$ is lower bounded by a constant times $r^{d}$ as $r \to 0$ (using that $\mathcal{X}$ is bounded).
> Note, however, that Assumption 4 is not always satisfied by convex sets with an empty interior. For instance, if $d=2$ and the domain $\mathcal{X}$ is a one-dimensional segment, then Assumption 4 does not hold. In fact, the role of Assumption 4 is to make $d$-dimensional integration on $\mathcal{X}$ meaningful, in particular excluding cases where $\mathcal{X}$ is a manifold of dimension strictly less than $d$ embedded in $\mathbb{R}^d$.
> We will add a little more intuition on Assumption 4 in the revised version.
>
> 5. Thank you for catching it! We accidentally overloaded the notation $K$ without realizing it. We will make sure to update it in the revised version.
>
> 6. *Is it possible that the exact knowledge of the Lipschitz constant can be used to get an improved algorithm for any function f?* Good question! In Proposition 2, we only showed that the Piyasvskii-Shubert algorithm improves on the bound of Proposition 1 for the special function $f(\cdot) = L || \cdot ||$ in dimension $d \geq 2$, and the question is still open for any other function $f$. We believe the Piyasvskii-Shubert algorithm could also feature an improved sample complexity for some other functions $f$, but that there are some functions $f$ for which knowing $\mathrm{Lip}(f)$ exactly is not helpful. For example, we conjecture that the upper bound of Proposition 1 is optimal up to a log term for functions $f$ that have relative variations $|f(x)-f(y)|/||x-y||$ equal to $\mathrm{Lip}(f)$ far away from the set of maximizers but that are $L'$-Lipschitz near the set of maximizers, with $L' < \mathrm{Lip}(f)$.
>
> ---
>
> We thank the reviewer again for their insightful comments, whose answers will improve the quality of the paper.

---

> > ### Comment · Reviewer_FGKi · 2021-08-31
> > **Thanks**
> >
> > Thanks a lot for carefully answering my questions. I believe that incorporating these answers to the paper will improve the presentation of the paper.

---

### Official Review · Reviewer_wYFu · 2021-07-24

**Rating:** 7
**Confidence:** 3

**Summary:**

The authors introduced the Certified DOO algorithm, which is a zero-order method, to find the maximal point of a Lipschitz function on a compact feasible set. The authors also provided the convergence rate of the Certified DOO algorithm and an instance-dependent lower bound for all certified algorithms. And the lower bound matches the upper bound op to a logarithm term.

**Limitations And Societal Impact:**

Yes.

**Main Review:**

The paper is well-organized. The authors present the Certified DOO algorithm and the upper bound of sample complexity of the Certified DOO algorithm. And the novel technique, which the authors used to prove an instance-dependent lower bound of certified algorithms, could likely follow up to prove other learning settings.  With noticing that the Certified DOO algorithm is highly dependent on the information of the feasible set, there is a question that I am interested in: can we design algorithms without knowing the information of the feasible set $K$ in advance, i.e., the oracle will return $- \infty$ if $x \notin K$.

Thanks for the authors' nice response. I would like to keep my ratings for this good paper,

**Time Spent Reviewing:**

about 8 hours.

---

> ### Author Response · Authors · 2021-08-10
> **Response to Reviewer wYFu**
>
> We thank the reviewer for the time spent reading our paper and for sharing their comments.
> We will update the revised version in light of your feedback.
>
> ---
>
> It is true that (not only the certified DOO algorithm but) most optimization algorithms are highly dependent on the information on the feasible set. It is indeed a standard assumption that the learner knows the feasible set $K$ (or $\mathcal{X}$, with the paper's notation) in advance.
>
> That said, we can certainly consider the design of algorithms in the case where $K$ is not known and the oracle returns $- \infty$ if $x \notin K$.  In the Bayesian optimization community, these algorithms already exist (in the non-certified case). Typically they rely on both a Gaussian process model for the Lipschitz function on $K$ and a classifier to learn whether $x \in K$ (feasible point) or not (unfeasible point). The next query point is found by maximizing an acquisition function based on a combination of the expected function value (if the point was feasible) and the probability of being feasible. We refer for instance to D. V. Lindberg and H. K. H. Lee "Optimization Under Constraints by Applying an Asymmetric Entropy Measure", Journal of Computational and Graphical Statistics.
>
> We also think that the (non-certified) DOO algorithm could be extended to the case where $K$ is not known but is contained in a known bounded set (note that if the latter is false, the problem becomes hopeless). We expect that a successful adaptation would be to alternate between the two following choices of the next cell to explore. (1) The choice of the paper in Algorithm 1, line 5, for which the explored cell will have a feasible representative. (2) If there are cells with unfeasible representatives, explore the one with smaller depth $h$ (breaking ties arbitrarily). Under appropriate regularity assumptions on the feasible set $K$, it is an interesting open problem to obtain sample complexity bounds on this extended algorithm. A subsequent open problem is then to add a certificate to this algorithm and again study its certified sample complexity.
>
> We thank the reviewer for this question. We are happy to add it to the open problems section and investigate it further in the future.

---

### Official Review · Reviewer_Ri1i · 2021-07-27

**Rating:** 7
**Confidence:** 4

**Summary:**

The paper studies the instance-dependent sample complexity for zeroth-order optimization of a Lipschitz function f, with the additional constraint that the algorithm must certify the accuracy of the output. The paper provides nearly matching instance-dependent upper and lower bounds for the setting of perfect (deterministic) observation.


**Limitations And Societal Impact:**

The authors have adequately addressed the limitations and potential negative societal impact of their work.

**Main Review:**

Overall, I believe this work is a nice complement to the instance-dependent sample complexity of zeroth-order optimization. The presentation is good with clear upper and lower bounds. My main concerns are summarized as below.

1. I'm confused about the gap in the dependence of dimension $d$ in the upper and lower bounds. Theorem 3 suggests that the constant $c$ in lower bound is $c'(1-Lip(f)/L)^d$, and claims that the results are direct corollary of Theorem 2 and Proposition 1. However, in theorem 2 the constant $c$ has a extra dependence of $2^{-7d}$, which can be huge when the dimension is large and suggests another mismatch of order $2^{-d}$ in the upper and lower bounds. Is there any other technique applied to mitigate this factor in the lower bound? Otherwise, the authors shall state this gap clearly in the introduction and theorem instead of hiding it in the constant $c'$.

2. Based on the dependence of smoothness in the lower bound in Theorem 2, the adaptivity to all range of smoothness seems to be a natural and important question towards fully instance-dependent optimal bound. It is mentioned in line 89 that 'our lower bound suggests that no adaptivity could be possible for certified algorithm'. This is not clear from Proposition 2 and 3, since Proposition 2 seems to suggest that there exists algorithm that can achieve constant sample size when $f$ is exactly $L$-Lipschitz. Could the author elaborate on this point?

3. The mismatch between one dimension and high dimension is quite interesting. In Proposition 3, the authors suggest that in one dimension, the term $S_c$ is again a valid lower bound. Does this mean that the problem when we know exactly the smoothness of the function in one dimension can be harder than that in high dimension? Does this also imply that there exists algorithm that is adaptive to smoothness in one dimension? It would be great if some intuitive explanation can be added on why the discrepancy between one dimension and high dimension happens.

Besides the concerns mentioned above, I think understanding exactly the instance-dependent optimal zeroth-order optimization is a good contribution. I'm happy to adjust the points accordingly if my questions are resolved.


***Update based on the author response:
The authors have resolved most of the questions listed above. Although the exponential gap in dimension is still tricky, I acknowledge that this is an open problem left and this work makes a first step towards understanding the instance-dependent optimal zeroth-order optimization in high dimension. I have adjusted my score accordingly.


**Time Spent Reviewing:**

5

---

> ### Author Response · Authors · 2021-08-10
> **Response to Reviewer Ri1i**
>
> We thank the reviewer for the time spent reading our paper and for sharing their comments.
> We will update the revised version in light of your feedback.
>
> ---
>
> 1.  *Constants depending exponentially on $d$.* In this paper, we admittedly did not focus too much on finding the tightest possible multiplicative constants. As such, one could certainly expect to improve them even by simply applying more accurately our very same arguments. However, we believe that eliminating the exponential mismatch of upper and lower bounds altogether would be extremely challenging. This problem is common when working in high-dimensional spaces: any looseness in the analysis can propagate in $d$ dimensions, giving, in turn, an exponential deterioration of the constants. These considerations do not at all dismiss the reviewer's question. On the contrary, this is the next big open problem in certified optimization!
> To be more transparent, the dependency on $d$ will appear explicitly in the constants of all our results. For instance, in the informal theorem of the introduction as well as in Theorem 3, we will replace $c, c', C$ with $c_d, c'_d, C_d$ and provide their expressions.
> On a similar note, thanks to a suggestion from reviewer TJAZ, we will point out that $\gamma$ in Assumption $4$ also depends on $d$ and we will clarify why $L^d$ appears as a multiplicative constant in Theorem 1.
> We will include these remarks and add to the open problems the question of decreasing the gap between the multiplicative constants of the upper and lower bounds.
> We thank again you and the other reviewers for pointing this out!
>
> 2. *Adaptivity to smoothness.*
> We stated in line 89 that "our lower bound suggests that no adaptivity could be possible for certified algorithms", and the Reviewer raises a possible contradiction with Proposition 2. We think that the reviewer meant adaptivity to $\mathrm{Lip}(f)$ when an upper bound $L$ is known, while what we had in mind in line 89 was: adaptivity to $\mathrm{Lip}(f)$ when *no upper bound $L$ is available*. We provide details below.
> In Proposition 2 we showed that, in dimension $d \geq 2$ and for the specific function $f(\cdot) = L || \cdot ||$, the Piyavskii-Shubert algorithm tuned with $L$ attains the lower bound of Theorem 2 (up to an additive constant) and can thus adapt to the ratio $\mathrm{Lip}(f)/L$ being equal to $1$ for the special function $f(\cdot) = L || \cdot ||$. Completing the picture for *any* function $f$ satisfying $\mathrm{Lip}(f)=L$ (in dimension $d \geq 2$) is still an open question. We will state it explicitly in the open problems section. We would however like to stress that this (theoretically interesting) case has limited practical impact, since the learner would have to be very lucky to choose $L=\mathrm{Lip}(f)$ without knowing $f$.
> For example, this is very unlikely to happen in safety-critical applications, where $L$ should be chosen in a conservative fashion.
> What we meant by "adaptivity" is adaptivity to $\mathrm{Lip}(f)$ when no upper bound $L$ is available, which would have a more practical impact if that were possible. More precisely:
>
> (a) does it exist an algorithm that does not take the parameter $L$ as input but whose sample complexity could be upper bounded by a term of the order of
> $$\mathcal{N} \left( \mathcal X_\varepsilon, \frac{\varepsilon}{\mathrm{Lip}(f)} \right) + \sum_{k=1}^{m_{\varepsilon}} \mathcal{N} \left( \mathcal X_{(\varepsilon_k, \varepsilon_{k-1}]}, \frac{\varepsilon_k}{\mathrm{Lip}(f)} \right) ?$$
> In other words, are there algorithms that could reach the bound of Proposition 1 without requiring the knowledge of an upper bound $L$ on $\mathrm{Lip}(f)$?
>
> (b) alternatively, does it exist an algorithm taking $L$ as input and whose sample complexity would behave as in (a) above with an additional polylogarithmic dependency on $L/\mathrm{Lip}(f)$ (so that in practice, we could take a large $L$ and be confident that it is a valid upper bound on $\mathrm{Lip}(f)$, without affecting the performance of the algorithm too much)?
>
> Unfortunately, (a) and even the easier (b) are impossible except in trivial cases (e.g., when $\mathcal{X}$ is finite). Indeed, it is impossible to produce a certificate $\xi_n$ on the optimization error without knowing in advance how large the variations of $f$ can be. Another way to see this is by using the lower bound of Theorem 2. For example, by letting $L \to +\infty$ in Theorem 2 (the algorithm is certified for any $L$ if (a) is true), the lower bound is proportional (up to a logarithmic factor) to  $S_{\mathrm{C}}(f,\varepsilon) \geq \mathcal{N} \left( \mathcal{X}_{\varepsilon}, \frac{\varepsilon}{L} \right) \ge \text{const} \left( \frac{L}{\mathrm{Lip}(f)}\right)^d \to +\infty$ whenever there is a maximizer of $f$ in the interior of $\mathcal{X}$. A similar analysis can be made for disproving (b).
> This explains why we suggested that "no adaptivity could be possible".
> Using a doubling-trick (by splitting the query budget into different periods where we tune $L$ with geometrically increasing values), it might however be possible to obtain a weaker version of (b), by outputting at every round $n$ a family of pairs (recommendation, error certificate), one of which would be valid. This would enable the practitioner to think of an upper bound $L$ on $\mathrm{Lip}(f)$ only after running the algorithm, instead of before running it. We will make sure to clarify our comment on adaptivity and to mention this last question as an open problem. We thank the reviewer for pointing this out.
>
> 3. *Is the problem harder in dimension $1$ than in higher dimensions when $L=\mathrm{Lip}(f)$?*
> Propositions 2 and 3 do not go against the intuition that certification in dimension $d \geq 2$ is harder than in dimension $1$ when the best Lipschitz constant of $f$ is known exactly. Indeed, even if Proposition 3 shows that $S_{\mathrm{C}}(f,\varepsilon)/(1+m_{\varepsilon})$ is a valid lower bound up to constants in dimension $1$ (while it is not always the case in dimensions $d \geq 2$, cf. the counterexample of Proposition 2), this lower bound is actually of the order of a constant for the special function $f(x) = L |x|$. Therefore, this does not imply that the problem is hard in dimension $1$ for this function.
> The message of the two propositions is that $S_{\mathrm{C}}(f,\varepsilon)$ is the right quantity, that fully and tightly (up to $\log(1/\varepsilon)$) characterizes certified optimization in one dimension. In dimension $d > 1$, this is also true when $\mathrm{Lip}(f)$ is bounded away from $L$, but in the (quite unrealistic) case when $\mathrm{Lip}(f)$ is known exactly and we choose $L=\mathrm{Lip}(f)$, then for some very specific functions (e.g., norms), the quantity $S_{\mathrm{C}}(f,\varepsilon)$ can be loose.
> We will make sure to clarify this as some other reviewers had similar questions as well.
> Finally, let us provide some intuition on the discrepancy between dimension $1$ and $d \geq 2$. The quantity $S_C(f , \varepsilon)$ is a nearly-tight lower bound in dimension $1$ but not in dimension $d \geq 2$. The intuition that we have is the following. When $S_C(f, \varepsilon)$ is large, then there are many separated points that are near-optimal for $f$. This yields disjoint balls where we can locally change $f$ to "trap" an algorithm when $n$ is much smaller than $S_C(f, \varepsilon)$ (which is what happens in the proofs of Theorem 2 and Proposition 3).
> In dimension $1$ we put a triangle of maximum slope $L$ at any of these balls (which is what happens in the proof of Proposition 3). This is because, in dimension $1$, when we modify the values of a Lipschitz function on a ball, we only have to satisfy continuity on the two endpoints of this ball. However, in dimension $d \geq 2$, we can not always put a "pyramid" of maximum slope $L$ at these balls. Indeed, modifying the values of a Lipschitz function on a ball is harder in dimension $d \geq 2$, as there are now continuity constraints on an uncountable set (a sphere) rather than on two points. It is for this reason that the proof of Theorem 2 needs $L < \mathrm{Lip}(f)$. We will add this discussion to the paper.
> 4. *Does it exist an algorithm that is adaptive to smoothness in one dimension?*
> If the question is in terms of adaptivity to $\mathrm{Lip}(f)$ when an upper bound $L$ is known, then we know from Propositions 1 and 3 that the optimal sample complexity in dimension $d=1$ is given by $S_C(f, \varepsilon)$ up to a logarithmic term that *does not* involve the ratio $\mathrm{Lip}(f)/L$. The question of being adaptive to $\mathrm{Lip}(f)$ when an upper bound $L$ is known thus does not really arise in dimension $1$. To put it differently, the c.DOO algorithm is (immediately) adaptive to $\mathrm{Lip}(f)$ in dimension $1$ when an upper bound $L$ is known.
>
> ---
>
> We thank again the reviewer for their comments and suggestions. We are happy to answer any other follow-up questions (if any) during the discussion period. If all doubts were resolved, we would be happy if the reviewer could consider raising their score.

---

### Official Review · Reviewer_8oRd · 2021-07-30

**Rating:** 6
**Confidence:** 4

**Summary:**

This paper presents tight, instance dependent upper and lower bounds for certifiable zeroth-order Lipschitz function optimization. It was known from prior work that in 1-d, the optimal sample complexity is nearly proportional to the integral $\int_{\mathcal{X}} \frac{\mathrm{d} x}{\max (f) - f (x) + \epsilon}$.  This paper extends this result to dimension $d$, establishing that the DOO algorithm of Perevozchikov 1990 can be made certifiable, achieving the sample complexity $\int_{\mathcal{X}} \frac{\mathrm{d} x}{(\max (f) - f (x) + \epsilon)^d}$.

The sample complexity bound of this algorithm matches that derived by Bouttier et al (2020) for an algorithm of Piyavskii and Shubert. The bound in Bouttier et al, however is provided in terms of covering numbers of the ground set at different error scales.

The main contribution of this paper is to (i) show that the c.DOO algorithm runs in near linear time and (ii) a sample complexity bound with of the same order as that derived by Bouttier et al (ii) To relate this sample complexity bound to the quantity $\int_{\mathcal{X}} \frac{\mathrm{d} x}{(\max (f) - f (x) + \epsilon)^d}$ (iii) establish a lower bound for certifiable algorithms which matches the upper bound up to an order of $(c( 1 - \text{Lip} (f) / L ))^d$ where $L$ is a known upper bound on the Lipschitz constant of the true function.

The authors also complement these results with a sample complexity upper bound of a constant when $L = \text{Lip}(f)$ (i.e. the Lipschitz constant of the function is known) for $d \ge 2$, which surprisingly does not hold for the case of $d=1$.

**Ethical Concerns:**

None to the best of my knowledge.

**Limitations And Societal Impact:**

Yes, I believe the authors have. It might help to include some pointers to questions left open in this work in the conclusion section.

**Main Review:**

While I think the contribution of the paper is interesting, I think that this result should be advertised less as resolving the open problem of finding the number of sample complexity queries of any optimal algorithm. Indeed it should be made more explicit that: (i) the authors relate the sample complexity bound on the PS algorithm established earlier by Bouttier et al, and (ii) show that the DOO algorithm can be made certifiable. I feel like these points are not established clearly in the abstract. I provide my remaining comments below:

* It should be mentioned that the constants c and C in the upper and lower bounds depend exponentially on d and are not absolute constants.

* Theorem 2 of Bouttier et al (2020) proves that the term $S_C (f,\epsilon)$ is an upper bound to the sample complexity for a different algorithm than the one considered in the paper which is less computationally efficient. It might help to clarify this again in the discussion above eq. (5) / in section 2.

* In the intuition for proposition 2, I find it hard to understand how the learner can *find* two points at which the respective values of f are maximally distant. Even given two such points, the general intuition for this result is not clear to me.

* I find it hard to understand the message in propositions 2 and 3. Shouldn’t the certification problem in, say, 2 dimensions be harder than in 1 dimension? One can always embed a 1-d Lipschitz function into 2-d as: $f_2 ( x ) = f_1 ( \langle x , e_1 \rangle )$.

* Is there any intuition as to what to expect in the case when any of the assumptions such as 2 and 3 are relaxed? Is there a fundamental difficulty in the absence of either of these assumptions?

**Time Spent Reviewing:**

4

---

> ### Author Response · Authors · 2021-08-09
> **Response to Reviewer 8oRd**
>
> We thank the reviewer for the time spent reading our paper and for sharing their comments.
> We will update the revised version in light of your feedback. We try below to address each of your concerns or questions.
>
> ---
>
> $\bullet$
> We will make the two contributions (i) and (ii) pointed out by the reviewer (lines 2 and 3 after "Main review") more explicit, in particular in the abstract. We thank the reviewer for the suggestion.
>
> $\bullet$
> As suggested, we will mention explicitly that the constants $c$ and $C$ depend exponentially on $d$. We thank you and the other reviewers that noticed it for catching that we were not always consistent in giving out explicit constants (which is what we usually prefer doing).
>
> $\bullet$
> As suggested, we will clarify in Section 2 that the bound of Bouttier et al (2020) addresses an algorithm that is less computationally efficient than the one we consider. This will complement the discussion about computational efficiency already present after eq. (5) in Section 1.3.
>
> $\bullet$
> As noted by the reviewer, the intuition behind Proposition 2 is that if there are two points at which the respective values of $f$ are maximally distant, then this can help the learner.
> This is what happens in the proof of Proposition 2, where the first two query points $x_1$ and $x_2$ enable to have an error certificate that is exactly zero. Indeed, since in our case $x_1$ is the center of the unit ball and $x_2$ is on the unit sphere, the distance between $x_1$ and any point $u$ of the ball is smaller or equal to the distance between $x_1$ and $x_2$, so that, by the Lipschitz property,  $f(u) \leq f(x_1) + L ||u-x_1|| \leq f(x_1) + L ||x_2-x_1|| = f(x_2)$. This guarantees that $x_2$ is a maximizer of $f$.
>
> A simpler example that brings some more intuition in dimension $1$ is when the domain is $[0,1]$, $L=1$, and we observe $f(0) = 0$ and $f(1) = 1$. Then we know that the $1$-Lipschitz function is the identity on $[0,1]$, from only these two observations.
>
> However, Proposition 2 and this one-dimensional example are cases where a specific initialization of the algorithm was used to yield these two points with maximally distant values.
> As such, this is merely a tool to study $f$-dependent lower bounds (in particular, to provide counterexamples).
> In practice, it seems hard to design an algorithm that reliably finds two such points.
>
> $\bullet$ About the message of Propositions 2 and 3. We would first like to add a clarification to what is stated in the review: the sample complexity upper bound of a constant when $L = \mathrm{Lip}(f)$ and $d \geq 2$ (Proposition 2) is shown only for the specific function $f(\cdot) = L || \cdot ||$ and not for all functions. Besides, the same constant upper bound can also be proved for the function $f(\cdot) = L | \cdot |$ when $d=1$ (in this case, the lower bound of Proposition 3 is also constant). Therefore, the general message behind Propositions 2 and 3 is not to show that the certification problem is harder in dimension 1 than in dimensions 2 or larger, but rather:
>
> -- to provide a counterexample showing that $S_C(f , \varepsilon)$ is not a tight lower bound in general (even up to the factor $\log(1/\varepsilon)$) when $L = \mathrm{Lip}(f)$ in dimension $d \geq 2$, as shown by Proposition 2.
>
> -- to show that, in dimension $d=1$, $S_C(f , \varepsilon)$ is however a nearly-tight lower bound for any $f$ even when $L = \mathrm{Lip}(f)$, as stated in Proposition 3.
>
> In other words, the message of the two propositions is that $S_{\mathrm{C}}(f,\varepsilon)$ is the right quantity, that fully and tightly (up to $\log(1/\varepsilon)$) characterizes certified optimization in one dimension.
> In dimension $d > 1$, this is also true when $\mathrm{Lip}(f)$ is bounded away from $L$, but in the (quite unrealistic) case when $\mathrm{Lip}(f)$ is known exactly and we choose $L=\mathrm{Lip}(f)$, then for some very specific functions (e.g., norms), the quantity $S_{\mathrm{C}}(f,\varepsilon)$ can be loose. We will more explicitly state this as an open question. We will also clarify the discussion of Propositions 2 and 3 with the above comments.
>
>
> $\bullet$ Assumptions 2 and 3 are on the infinite tree that regulates the behavior of the certified DOO algorithm.
> They make it so that the cells become smaller as the depth $h$ increases and that we cannot find distinct cell centers that are too close to each other.
> These mild assumptions are designed for efficient use of the function query budget by the infinite tree of DOO. Intuitively, if these assumptions were removed altogether, this would allow for very inefficient choices of the infinite tree for which no optimal sample complexity bound would hold. An extreme example is if a sequence of cells of depth $h=1,h=2,\ldots$ contains a fixed ball centered at the global maximizer. Then the DOO algorithm would not even be provably asymptotically consistent.
> That said, we do not know if the two assumptions are entirely necessary, or if it would be possible to consider more general trees whilst retaining the same performance.
>
> $\bullet$
> As suggested, we will add pointers to open questions in the conclusion section, including in particular all of the points raised by the reviewers.
>
> ---
>
> We thank again the reviewer for their comments and suggestions. We are happy to answer any other follow-up questions (if any) during the discussion period. If all doubts were resolved, we would be happy if the reviewer could consider raising their score.

---

> > ### Comment · Reviewer_8oRd · 2021-08-20
> > **Response to the authors**
> >
> > I thank the author for the responses. The clarification of the contribution of propositions 2 and 3 makes sense. Somehow reading through the paper, I did not catch this. Perhaps it might help to clarify this by including the two points explicitly at the start of the section.
> >
> > However, as is also pointed out by reviewer QEpC, I maintain my position that a majority of the heavy lifting is carried out in the paper by the parameter $S_C (f,\epsilon)$ introduced by Bouttier et al (2020). With this, I keep my score for the paper unchanged.

---

### Author Response · Authors · 2021-08-10
**TL;DR of the responses**

We are thrilled by the positive feedback and strong interest expressed by the reviewers but, in writing the seven responses, we realized that it could be a bit labyrinthic to get through all of them, especially given that most suggestions (and therefore, answers) appear multiple times. We thought it could be helpful to sum things up.

Beyond some very minor individual suggestions (a language improvement here, an additional explanatory sentence there), the reviewers requested three changes.

1) Highlight the dependency on the dimension $d$ of some constants (in the few statements in which we did not already).

2) Move to a separate (sub)section the boundary case in which $\mathrm{Lip}(f)=L$ is known exactly, and add a short paragraph to give a bit more intuition on these results.

3) Move the "open problems" paragraph of lines 85-91 to a (sub)section at the end, and add a few more suggestions that came from the reviewers.

We agree with the reviewers that with these minor changes, the overall clarity of the paper will improve, and we are happy to commit to it.

---

### Decision · Program_Chairs · 2021-09-27

**Decision:**

Accept (Poster)

**Comment:**

This paper considers the zeroth-order optimization problem with error certificates for Lipschitz functions. It generalized existing work to higher dimensions and showed that the algorithm is certifiable. The technical contribution is solid. I am happy to recommend acceptance, but I also note that the reviewers have pointed out that the technical novelty is limited in the sense that it owes a lot to Bouttier et al. (2020) which obtained the crucial quantity S_C(f,\varepsilon).